# Decreased Tissue Omega-6/Omega-3 Fatty Acid Ratio Prevents Chemotherapy-Induced Gastrointestinal Toxicity Associated with Alterations of Gut Microbiome

**DOI:** 10.3390/ijms23105332

**Published:** 2022-05-10

**Authors:** Kanakaraju Kaliannan, Shane O. Donnell, Kiera Murphy, Catherine Stanton, Chao Kang, Bin Wang, Xiang-Yong Li, Atul K. Bhan, Jing X. Kang

**Affiliations:** 1Laboratory for Lipid Medicine and Technology, Department of Medicine, Massachusetts General Hospital and Harvard Medical School, Boston, MA 02129, USA; kanakaraj.revathi@gmail.com (K.K.); yy_bwang@hotmail.com (B.W.); xyli75@126.com (X.-Y.L.); 2School of Microbiology, University College Cork, T12 K8AF Cork, Ireland; odonnell.shane22@gmail.com (S.O.D.); catherine.stanton@teagasc.ie (C.S.); 3Teagasc Moorepark Food Research Centre, Fermoy, P61 C996 Co. Cork, Ireland; 4APC Microbiome Ireland, University College Cork, T12 YT20 Cork, Ireland; kiera.healy@teagasc.ie; 5Department of Nutrition, The General Hospital of Western Theater Command, Chengdu 610000, China; chao.kang_tmmu@hotmail.com; 6Department of Pathology, Massachusetts General Hospital and Harvard Medical School, Boston, MA 02114, USA; abhan@mgh.harvard.edu

**Keywords:** CPT-11, irinotecan, gut microbiome, FAT-1 transgenic mouse, omega-3 fatty acids, omega-6/omega-3 PUFA ratio, beta-glucuronidase, GUSB, chemotherapy-induced gut-toxicity

## Abstract

Gastrointestinal toxicity (GIT) is a debilitating side effect of Irinotecan (CPT-11) and limits its clinical utility. Gut dysbiosis has been shown to mediate this side effect of CPT-11 by increasing gut bacterial β-glucuronidase (GUSB) activity and impairing the intestinal mucosal barrier (IMB). We have recently shown the opposing effects of omega-6 (n-6) and omega-3 (n-3) polyunsaturated fatty acids (PUFA) on the gut microbiome. We hypothesized that elevated levels of tissue n-3 PUFA with a decreased n-6/n-3 PUFA ratio would reduce CPT-11-induced GIT and associated changes in the gut microbiome. Using a unique transgenic mouse (FAT-1) model combined with dietary supplementation experiments, we demonstrate that an elevated tissue n-3 PUFA status with a decreased n-6/n-3 PUFA ratio significantly reduces CPT-11-induced weight loss, bloody diarrhea, gut pathological changes, and mortality. Gut microbiome analysis by 16S rRNA gene sequencing and QIIME2 revealed that improvements in GIT were associated with the reduction in the CPT-11-induced increase in both GUSB-producing bacteria (e.g., *Enterobacteriaceae*) and GUSB enzyme activity, decrease in IMB-maintaining bacteria (e.g., *Bifidobacterium*), IMB dysfunction and systemic endotoxemia. These results uncover a host–microbiome interaction approach to the management of drug-induced gut toxicity. The prevention of CPT-11-induced gut microbiome changes by decreasing the tissue n-6/n-3 PUFA ratio could be a novel strategy to prevent chemotherapy-induced GIT.

## 1. Introduction

Chemotherapy-induced gastrointestinal toxicity (CIGIT) is a common side effect in cancer patients, reported as high as 50–80% in patients treated with certain drugs such as Irinotecan in the regimens [1]. Irinotecan or Camptothecin-11 (CPT-11), first-line chemotherapy for advanced colorectal cancer, causes delayed diarrhea in up to 87% of patients, with 30–40% suffering severe diarrhea (grade 3 or grade 4) [2]. CPT-11 is a prodrug hydrolyzed by carboxylesterases to the active metabolite SN-38, a topoisomerase I inhibitor [3]. SN-38 further undergoes hepatic glucuronidation very efficiently to form the inactive SN-38 glucuronide (SN-38G) by UDP-glucuronosyltransferase 1A1 [4], which is then excreted into the gastrointestinal (GI) tract via bile [5]. When SN-38G reaches the intestine, it is subjected to microbiota expressing β-glucuronidase (GUSB) that converts SN-38G back to SN-38 [6]. Increased exposure of intestinal epithelia to the released SN-38 causes histological abnormality, such as a net decrease in mucin-producing goblet cells [7,8], changes in mucin (MUC) expression and secretion [7,8] and bacterial invasion into the mucosa, and damage in the intestine, which is believed to be the leading cause of CPT-11-induced diarrhea [6]. 

The gut microbiota is an essential regulator of intestinal homeostasis, and alterations in microbial composition have been associated with multiple inflammatory diseases [9]. The gut microbiome plays a crucial role in CPT-11-induced diarrhea [9,10]. Brandi et al. observed that a 2.5-fold increased lethal dosage of CPT-11 in germ-free mice showed more resistance to CPT-11 than holoxenic mice [11]. Slatter et al. determined the pharmacokinetics of [14C] CPT-11 in human cancer patients and found that the SN-38 content increased from 0.44% of the administered dose in the bile to 2.78% in the feces [12]. A direct connection between irinotecan metabolism and the composition of an individual’s gut microbiota has recently been made in humans [13]. Specifically, the gut microbiota has GUSB activity, and the contribution of bacterial GUSB activity in CPT-11 toxicity has been well established [6,11,14]. The Gram-negative LPS-producing *Enterobacteriaceae* (e.g., *Escherichia coli*) are primary producers of GUSB. *Bacteroides* spp., *Staphylococcus* spp., *Streptococcus* spp., and *Clostridium* spp. have also been reported to produce GUSB [15,16]. Previous research has shown that some of these bacteria are altered in the GI tract after treatment with Irinotecan [16,17]. Of particular interest in this study are *E. coli*, *Bacteroides* spp., *Staphylococcus* spp., and *Clostridium* spp., all reported to produce GUSB. Also of interest are LPS-suppressing *Bifidobacterium* and *Akkermansia,* said to be beneficial and have protective properties towards the gut mucosal barrier [18,19]. Furthermore, data from a recent study have highlighted Toll-like receptor 4 (TLR4) involvement in developing irinotecan-induced gastrointestinal toxicities (GIT) [20]. In addition to lipopolysaccharides (LPS) from Gram-negative bacteria [20], SN-38 (the active metabolite of CPT-11) has the potential to act as a ligand for the TLR4 [20]. It is well known that TLR4-mediated signaling plays a central role in the development of irinotecan-induced intestinal barrier disruption, the elevation of gut permeability, and endotoxemia [20].

Currently, supportive methods such as antidiarrhea agents are used to relieve diarrhea symptoms, but the intestinal damage caused by CPT-11 persists [2]. Historically, oral antibiotics were used to reduce CPT-11-induced toxicity [21]; however, indiscriminate depletion of gut microbes may impair protective functions, including the ability to resist infection and the capacity to metabolize dietary substrates. Furthermore, gut microbiota depletion directly impacts chemotherapy treatment through a variety of mechanisms, including the prevention of beneficial crosstalk with the immune system [9]. Recent efforts to reduce CPT-11 toxicity include the targeted inhibition of microbial enzymes that convert the drug’s inactive form to its active form. Wallace et al., 2010, identified potent Escherichia coli BG inhibitors, which substantially reduce CPT-11-induced toxicity in mice while not affecting the orthologous mammalian enzyme [22]. Likewise, attempts to minimize the TLR4-mediated signaling consequently elevated GUSB-producing proteobacteria levels, increased the rate of SN- 38 reactivation, and worsened the CPT-11-induced GIT [20]. 

It is well recognized that long-chain omega-6 (n-6) and omega-3 (n-3) polyunsaturated fatty acids (PUFA) play essential and opposing roles in the modulation of inflammation [18,23]. We and others have recently demonstrated the opposing effects of n-6 and n-3 PUFA on the gut microbiome [18,24]. Reducing the tissue n-6/n-3 PUFA ratio with elevated tissue levels of n-3 PUFA can significantly reduce the growth of Gram-negative *Enterobacteriaceae* and increase beneficial Bifidobacterium growth and *Akkermansia* [18,24], reducing LPS production and thereby preventing gut barrier disruption and endotoxemia [18]. In this context, we hypothesize that the reduced tissue n-6/n-3 PUFA ratio with increased tissue levels of n-3 PUFA can prevent the changes in the gut microbiota and the occurrence of GIT and diarrhea caused by CPT-11 administration. To test this hypothesis, we used the FAT-1 transgenic mouse model [25] to determine the effects of reduced tissue n-6/n-3 PUFA ratio on CPT-11-induced changes in gut microbiota and GIT associated with diarrhea. The fat-1 transgenic mice carry a fat-1 gene encoding an n-3 fatty acid desaturase that can catalyze the conversion of n-6 to n-3 PUFA [25] and have high tissue levels of n-3 PUFA with a balanced n-6/n-3 PUFA ratio with no need for dietary manipulation. Compared with the conventional dietary intervention (which can introduce potential confounding factors), this genetic approach is more effective in balancing the n-6 to n-3 ratio because it not only elevates tissue concentrations of n-3 PUFA but also decreases the levels of excessive endogenous n-6 PUFA [25], which is ideal for identifying the specific roles of n-6/n-3 PUFA ratio and addressing host–gut microbiome interactions (without confounding impact of diet). Our findings reveal the preventive effects of a decreased tissue n-6/n-3 PUFA ratio on CPT-11-induced GIT by preventing the associated gut microbiome alterations, imbalances in the host–gut microbiome interactions, and impairment in the intestinal mucosal barrier (IMB) function induced by CPT-11 exposure.

## 2. Results

### 2.1. Decreased Tissue n-6/n-3 PUFA Ratio Reduces CPT-11-Induced Gut-Toxicity

Using an established mouse model (Appendix A) of CPT-11 exposure [22], we found that a decreased tissue n-6/n-3 PUFA ratio with elevated n-3 PUFA status (Appendix A) in the FAT-1 mice reduced the CPT-11-induced weight loss and late-onset diarrhea (Figure 1A–C). Although the FAT-1 mice exposed to CPT-11(FAT-1+CPT-11) were still significantly smaller than control, untreated Fat-1 animals, they were substantially more massive than WT mice exposed to CPT-11(WT+CPT-11) (Figure 1A). GI symptoms (diarrhea and bloody diarrhea) appeared after 6 days in the WT+CPT-11 group (Appendix A). On days 7–10, all the mice from the WT+CPT-11 group experienced diarrhea and bloody diarrhea (Figure 1B,C). By day 11, all the mice from the WT+CPT-11 group had to be euthanized (Figure 1B,C) when signs of moribund condition (Appendix A) were noticed. Conversely, the GI symptoms (diarrhea and bloody diarrhea) appeared after 8 days in the FAT-1+CPT-11 group, which experienced less diarrhea and bloody diarrhea (Figure 1B,C). Furthermore, the severity of diarrhea (Appendix A) on day 11 was markedly reduced in the FAT-1+CPT-11 group compared with the WT+CPT-11 group mice. Examination and scoring of intestinal tissue samples (cecum, proximal and distal colon) from each group showed that decreased n-6/n-3 ratio protected the mouse large intestine from CPT-11–induced damages, such as loss of the mucosa’s structure and consequent reduction of its thickness; reduced length of the colon crypt; intense infiltration of inflammatory cells; and loss of cell differentiation (Figure 1D,E and Appendix A). Next, in a separate experiment, after mice were treated with CPT-11 following the 4-day dosing plan, CPT-11–induced lethality was determined over 3 weeks as described previously [26]. After i.p., injection of CPT-11 at 50 mg/kg for four consecutive days, all WT mice died between days 12 and 13, compared with FAT-1 mice with a survival rate of ~80% (Appendix A). These results suggest that a decreased tissue n-6/n-3 PUFA ratio plays a pivotal role in protecting the mice from CPT-11–induced toxicities.

### 2.2. Decreased Tissue n-6/n-3 Ratio Reduces CPT-11-Induced Gut Microbiome Alterations 

Using V3–V4 16S rRNA amplicon sequencing of cecal contents, we found that the overall gut microbiome of CPT-11 treated FAT-1 mice with a lower tissue n-6/n-3 ratio was distinct from other groups regarding β (Figure 2A; PERMANOVA: *p* < 0.001) and α diversity (Figure 2B and Appendix A) measures. Then, the co-occurrence and antioccurrence patterns of taxa (Figure 2C), identified using stringent network analysis SparCC [27] and critical taxa (Figure 2D) and differentiating the four groups were established. At the phylum level, CPT-treated WT mice’s cecal contents exhibited a dramatic increase in Proteobacteria levels, up to 25%, and decreases in Bacteroidetes (RA = 32%) compared to vehicle-treated WT mice (Figure 2E). In the CPT-11 treated FAT-1 mice, however, the expansion of Proteobacteria was markedly reduced, down to only 2.9%, while maintaining vehicle treatment levels of Bacteroidetes (43%) (Figure 2E). The family-level taxonomic changes reveal that the Proteobacterial reduction observed in the FAT-1+CPT-11 group arose from the GUSB-producing Enterobacteriaceae (Figure 2F–H). The genus-level analysis showed GUSB-producing *Enterococcus* [22] (Figure 2I) and beneficial taxa involved in gut health (*Bifidobacterium* and *Akkermansia*) [28,29] (Figure 2J,K) as a signature of WT+CPT-11 and FAT-1+CPT-11 groups, respectively. Moreover, we identified co-occurrences (positive correlations) between several GUSB-producing taxa (indicated with *****), which is high in the WT+CPT-11 group, and antioccurrences (negative correlations) between the GUSB-producing taxa [16] and the gut-health [28,29] related beneficial taxa (indicated with #), which is high in the FAT-1+CPT-11 group (Figure 2C,D). A q-PCR-based analysis confirmed the higher RA of the total GUSB-producing taxa in the WT+CPT-11 group’s stool compared to FAT-1+CPT-11 group (Figure 2L). Interestingly, we found lower GUSB activities in the stool and cecal contents (Figure 2M) of the FAT-1+CPT-11 group on days 6 and 11, respectively, compared to the WT+CPT-11 group. However, no difference was observed between WT and FAT-1 groups with baseline activity measurements taken before CPT-11 treatment (Figure 2M). Furthermore, we found a lower GUSB gene expression with immunohistochemical staining of the colon (Figure 2N and Appendix A) and decreased predicted proportional abundance of the GUSB enzyme orthologs [Kyoto Encyclopedia of Genes and Genomes (KEGG) Orthology K01195, KEGG Enzyme EC 3.2.1.31] with PICRUSt2 (Phylogenetic Investigation of Communities using Reconstruction of Unobserved States2) (Figure 2O). Together, these findings indicate that decreased tissue n-6/n-3 ratio can significantly suppress CPT-11 induced increase in GUSB-producing gut microbiome and related GUSB activity.

### 2.3. Decreased n-6/n-3 Ratio Prevents CPT-11-Induced Intestinal Mucosal Barrier Dysfunction

We found that a decreased n-6/n-3 ratio in the FAT-1 mice treated with CPT-11 was associated with a reduction in bacterial invasion into the mucosal epithelium as measured by Gram staining (Figure 3A) and down-regulated toll-like receptor 4 (TLR 4) expression (Figure 3B), compared to WT+CPT-11 group. Furthermore, the FAT-1+CPT-11 group showed a decrease in markers of inflammation such as nuclear factor NF-kappa-B p65 subunit expression (Figure 3C), TNF-α (Figure 3D), IL-1β, IL-6, and MCP-1 (Appendix A)] and an increase in the marker of anti-inflammation [IL-10 (Appendix A)]. In addition, the reduced bacterial invasion and inflammation in FAT-1+CPT-11 mice were associated with improvements in the markers of colonic barrier integrity such as zonulin 1 (ZO-1) (Figure 3E), occludin, and claudin-1tight junction protein expressions (Appendix A), as well as serum LPS levels (Appendix A), compared to WT+CPT-11 group. This reduction in the mucosal inflammation was further confirmed by lower infiltration of neutrophils within the lamina propria mucosa of FAT-1+CPT-11, as measured by myeloperoxidase (MPO) [30] immunohistochemical staining (Figure 3F). Furthermore, the number of neutrophils present in colonic tissues was correlated with both the severity of inflammation (Figure 1D) and the histological score in these mice (Figure 1E). Likewise, the CPT-11-induced decrease in goblet cell numbers (determined by Alcian blue) and function (examined by Alcian Blue-acid mucins/PAS-acid and neutral mucins staining and mRNA expression analysis of MUC2 and MUC4) was prevented in the FAT-1+CPT-11 group compared to WT+CPT-11 group (Figure 3G–J). Together, these results indicate that a decreased n-6/n-3 PUFA ratio prevents CPT-11-induced intestinal barrier dysfunction measured by bacterial invasion markers, mucosal inflammation, and gut barrier and goblet cell dysfunctions.

### 2.4. Omega-3 PUFA Supplementation Reduces CPT-11-Induced Gut-Toxicity

To investigate the translational possibility of the beneficial findings observed in the FAT-1 mice, we tested whether supplementing WT mice with n-3 PUFA before administering CPT-11 can produce effects similar to those found in the FAT-1 mice. WT mice maintained on a chow diet were switched to either an n-6 PUFA-enriched corn oil (CO) or an n-3 PUFA-enriched fish oil (FO) diet for two months. After n-3 PUFA supplementation, the FO group mice exhibited tissue essential fatty acid profiles comparable to those of the FAT-1 transgenic mice (Appendix A). 

By following the same CPT-11 treatment protocol we used in the FAT-1 mice experiment (Appendix A), both CO and FO groups received CPT-11 and were allowed to eat the corresponding diets until the end of this experiment. We found that the WT mice supplemented with FO showed decreased body weight loss (Figure 4A), diarrhea (Figure 4B) and bloody diarrhea episodes (Figure 4C) and diarrhea severity (Appendix A) and, reduced CPT-11-induced damages in the mucosal layer (Figure 4D) with improved histopathological injury score in the large intestine compared to WT mice supplemented with CO (Figure 4E–G). Conversely, 10 days of fish oil supplementation to WT followed by CPT-11 administration for 9 days could not prevent CPT-11-induced gut toxicities (Appendix A). Both groups were euthanized on day 11 because of severe body weight loss and bloody diarrhea episodes. 

Next, by 16S rRNA gene sequencing of cecal contents, we found that the overall gut microbiome of the CPT-11 treated FO group with a lower tissue n-6/n-3 ratio was distinct from CPT-11 treated CO group regarding β (Figure 5A) and α diversity (Figure 5B and Appendix A) measures. Then, the co-occurrence and antioccurrence patterns of taxa (Figure 5C) and key taxa (Figure 5D) differentiating the study groups showed several GUSB-producing taxa (e.g., *Escherichia*) and beneficial taxa involved in gut health (e.g., *Bifidobacterium*) as a signature of CO and FO groups, respectively. Like the FAT-1 mouse study, we identified co-occurrences between major GUSB-producing taxa, which is high in the CO group, and antioccurrences between the GUSB-producing taxa and the gut-health related beneficial taxa, which are high in the FO group (Figure 5C,D). At the phylum level, the CPT-treated CO group’s cecal contents exhibited a dramatic increase in Proteobacteria levels, up to 18%, and undetectable levels of Actinobacteria (Figure 5E). In the CPT-11 treated FO group; however, the expansion of Proteobacteria was markedly reduced, down to only 1.5% (*p* < 0.05), while showing the presence of Actinobacteria at significant levels (1.6%; *p* < 0.05) (Figure 5E). The family-level taxonomic changes reveal that the Proteobacterial reduction observed in the FO group arose from the GUSB-producing Enterobacteriaceae (Figure 5F–H). The genus-level taxonomic changes reveal that the Actinobacterial elevation observed in the FO group arose from the gut-healthy *Bifidobacterium* (Figure 5I) genus. A q-PCR-based analysis confirmed the higher RA of the total GUSB-producing taxa in the CO group’s stool compared to the FO group (Figure 5J). Interestingly, we found lower GUSB activities in the stool and cecal contents (Figure 5K) of the FO group on days 6 and 11, respectively, compared to the CO group. However, no difference was observed between CO and FO groups with baseline activity measurements taken before CPT-11 treatment (Figure 5K). Furthermore, we found a lower GUSB gene expression with immunohistochemical staining of the colon (Figure 5L and Appendix A) and decreased predicted proportional abundance of the GUSB enzyme orthologs [KEGG Orthology K01195] with PICRUSt2 (Figure 2M).

Fish oil supplementation is effective in reducing the CPT-11-induced measures of colonic mucosal barrier dysfunction such as bacterial invasion into the mucosal epithelium (Figure 6A), mucosal inflammation (Figure 6B–E and Appendix A), gut barrier impairment (Figure 6F and S3J–L) and goblet cells dysfunction (Figure 6G–J) as we found in the FAT-1 mice.

### 2.5. Host–Gut Microbiome Interactions Driven by Tissue n-6/n-3 PUFA Ratio Might Be Involved in CPT-11-Induced GIT

The RV coefficient showed an overall measure of association between host parameters and microbiome profile both in the FAT-1 mice (0.81; *p* = 0.001) and FO supplementation (0.76; *p* = 0.02) experiments. Host–microbiome interactions with network-based analytical approaches between CPT-11 treated WT and FAT-1 samples resulted in a correlation network consisting of significant (*p* < 0.05) positive (edges colored blue) associations of total levels of GUSB-producing taxa with colonic tissue n-6/n-3 PUFA ratio, cecal contents GUSB activity, bloody diarrhea, markers of mucosal injuries and inflammation and serum LPS levels and, negative associations (edges colored red) with bodyweight and gut barrier integrity (Figure 7A). Almost a similar correlation network was found between CPT-11 treated CO and FO samples in addition to significant (*p* < 0.05) negative associations of gut-healthy *Bifidobacterium* with parameters as mentioned earlier involved in CPT-11-induced GIT (Figure 7B). A functional relationship between the host and microbiome is suggested by the similarity between host phenotypes and ASV types. After grouping the samples into high tissue n-6/n-3 ratio (WT+CPT-11 and CO+CPT-11 samples together) and low tissue n-6/n-3 ratio (FAT-1+CPT-11 and FO+CPT-11 samples together) phenotypes, the host and microbiome data showed a high degree of concordance that was statistically significant in Monte Carlo simulations with a *p*-value of 0.03. Superimposed microbiome and host data were separated not only by n-6/n-3 PUFA status but also by ASV type in the multivariate multiple factor analysis (MFA) (Figure 7C) and PCA analysis (Figure 7D). Multivariate biomarker analysis performed on the combined host and microbiome data using a random forest machine learning algorithm ranked post-CPT-11 cecal contents *GUSB* activity as the top hit that was different between high n-6/n-3 ratio and low n-6/n-3 ratio phenotypes, followed by tissue n-6/n-3 PUFA ratio, *Escherichia*, *Enterococcus*, and tissue n-6 PUFA (Figure 7E), suggested their strong correlation with the GIT. Performance evaluation based on the area under the curve (AUC) values of Receiver Operator Characteristic (ROC) curve showed a 100% prediction accuracy between high and low n-6/n-3 ratio phenotypes (AUC = 0.94; *p* < 0.001) for combined host and microbiome data (Figure 7F) compared to biomarker analysis using only the host (AUC = 0.91; *p* < 0.01) or the microbiome data (AUC = 0.92; *p* < 0.01) alone. A classical univariate ROC curve analysis on individual biomarkers showed significant (*p* < 0.05) AUC values of 0.89, 0.88, 0.87, and 0.73 for post-CPT-11 cecal contents *GUSB* activity (Figure 7G,H), n-6/n-3 PUFA ratio, *Escherichia,* and baseline fecal *GUSB* activity, respectively. Furthermore, multivariate models of Cox Proportional-Hazards analysis showed that the risk for the diarrhea episodes significantly increases by 94 (Hazard ratio; *p* < 0.0001) and 4.4 (Hazard ratio; *p* < 0.003) for the subjects with high baseline tissue n-6/n-3 ratio and high baseline fecal GUSB activity, respectively.

In summary, elevated tissue omega-3 PUFA status with a decreased tissue n-6/n-3 PUFA ratio prevents CPT-11-induced GI toxicity associated with alterations of the gut microbiome (changes in gut microbiota composition and function). In addition to the anti-inflammatory and mucosal protective effects, a decreased tissue n-6/n-3 PUFA ratio reduces the abundance of GUSB-producing bacteria, GUSB activity, and potentially the conversion of inactive SN-38G to toxic SN-38 and increases the abundance of gut-healthy bacteria and the balance of host–microbiome interactions in the gut. These alterations together with other gut-microbiota-independent mechanisms lead to reduced mucosal injuries, mucosal inflammation, impairment in the gut barrier, and systemic endotoxemia, resulting in the prevention of CPT-11-induced gut toxicities (Figure 8).

## 3. Discussion

Acute, dose-limiting GI side effects frequently limit the use of CPT-11 as an anti-cancer drug. Efforts to relieve toxicity by formulating novel strategies and agents for prophylaxis, structural and chemical modification, and modifying drug delivery have not been successful [2]. Extensive studies have described the gut microbiome’s mechanistic role in the GIT, but microbiome-based nutritional intervention strategies are scarce. By utilizing a unique FAT-1 transgenic animal model free from typical confounding factors of diet affecting the gut microbiome [19,25,31], high throughput 16S rRNA gene amplicon sequencing with QIIME2 bioinformatics analysis, and multivariate HMI analyses, we demonstrate that the prevention of CPT-11-induced gut dysbiosis by elevated tissue n-3 PUFA status and decreased tissue n-6/n-3 PUFA ratio as a novel strategy to prevent CIGIT. Our data collectively showed that a decreased tissue n-6/n-3 ratio reduced weight loss, diarrhea, and bloody feces; increased survival times; and improved survival rates in the mice. This suggested that decreased tissue n-6/n-3 ratio could reduce short- and long-term CPT-11-induced clinical manifestations. The significant differences in clinical manifestations between FAT-1 and WT mice can be attributed to their respective tissue n-6 and n-3 PUFA composition as they were maintained on a single identical diet. As reported in the present study, Higuchi, et al. [32] showed similar opposing effects of n-6 and n-3 PUFA on the bodyweight loss and survival rate of mice in the context of Methotrexate-induced gut toxicities.

The gut microbiome plays a crucial role in maintaining intestinal homeostasis; it regulates metabolic enzymes and transporters to influence some drugs’ efficacy and toxicity [13,33,34]. When CPT-11 enters the body, it is transformed to SN-38 by carboxylesterase [6,35]. Inactive SN-38G, a glucuronic-acid conjugate of SN-38, is reactivated into toxic SN-38 via hepatic and intestinal circulation and by bacterial GUSB within the intestinal tract [6,35]. Several animal studies suggest that host factors [36] such as increased proinflammatory cytokines and alteration of gut microbiome-regulating factors play a key role in microbiome changes in response to CPT-11 administration. Briefly, CPT-11 triggers the innate immune response and key pro-inflammatory immune cells such as macrophages [37] and Th17 cells [38] to cause the secretion and release of proinflammatory cytokines such as interleukin (IL)-18, IL-1β, IL-6, and tumor necrosis factor-α [39,40]. The increase in the level of proinflammatory cytokines accelerates the discharge of mucin stored in goblet cells and thereby induces vacuole formation, which further influences intestinal microbial ecocline by reducing the number of adhesion sites and decreasing nutrition. These changes cause a reduction in the number of symbiotic bacteria (e.g., *Bifidobacterium* spp.) and an increase in the number of opportunistic pathogens (e.g., *Escherichia* coli) [40]. Likewise, the initial production of SN-38 by GUSP-producing commensal gut bacteria [41,42] or pro-inflammatory gut microbial environment [18] leads to changes in the host factors (e.g., Intestinal Alkaline Phosphatase (IAP) [18,43] that can regulate the gut microbiome composition). The CPT-11 metabolites-induced gut mucosal injury/inflammation further down-regulate IAP [44] activity and other gut-microbiome regulating host factors, which in turn, causes the gut Enterobacteriaceae expansion. A “vicious cycle” may be formed because CPT-11 might further aggravate this disturbed status instead of readdressing it. This inharmonious state has been called the microbiota–host–CPT-11 axis [45].

It is well-documented that CPT-11 influences the gut microbial community, increases GUSB-producing bacteria (e.g., Enterobacteriaceae) [46] and GUSB activity, decreases intestinal health bacteria (e.g., *Bifidobacterium* and *Akkermansia*), and leads to higher SN-38 accumulation in the intestinal lumen; this, in turn, induces intestinal epithelial cell injury and inflammation and causes late-onset diarrhea. In a recent study [46], CPT-11-treated mice exhibited a dramatic increase in *Proteobacteria* levels, up to 68%, and the *Proteobacterial* expansions observed with CPT-11 arose from the *Enterobacteriaceae*. Furthermore, the *Enterobacteriaceae* is the only intestinal taxa that encode a GUSB operon containing the GUSB gene and glucuronide transporters [47], which may give these relatively trace *Enterobacteriaceae* taxa the ability to outcompete the more abundant *Bacteroidetes* by increasing glucuronic acid utilization [46,47]. Notably, the *Enterobacteriaceae* only encode L1 GUSB enzymes, which process SN38-G most efficiently [47]. Hence, bacterial GUSB, produced by CPT-11 and elevated n-6/n-3 ratio-induced *Enterobacteriaceae* overgrowth, is considered a crucial contributor to GIT and delayed diarrhea. In our study, higher levels of total GUSB-producing taxa may be responsible for the increased expression of the bacterial GUSB gene and GUSB activity in the CPT-11 treated WT animals. Among them, the *family Enterobacteriaceae,* the primary GUSB-producers, were the major ones, contributing ~25% of the sum of RA. Conversely, the growth of these GUSB-producing taxa was significantly reduced, and the growth of commensal (e.g., Bacteroidetes) and intestinal health bacteria (e.g., *Bifidobacterium* and *Akkermansia*) were increased by a balanced n-6/n-3 ratio of FAT-1 mice mainly through the reduction in Proteobacteria (RA ~2.9%). These results are supported by the opposing effects of n-6 and n-3 PUFA on the gut microbiome composition and function (e.g., LPS biosynthesis), shown by several recent studies [18,19,48,49]. Notably, elevated tissue levels of n-3 PUFAs are effective in reducing proinflammatory cytokines [50,51,52] and upregulating the gut-microbiome-regulating factor IAP [18]. Our studies have shown an attenuation of colonic markers of inflammation (e.g., TNF-α) and a significant improvement of chemical-induced colitis in FAT-1 mice with elevated n-3 PUFA levels [23,53]. Furthermore, we have shown that elevated tissue n-3PUFAs enhance intestinal production and secretion of IAP [18], leading to changes in the gut bacteria composition with a decrease in the members of phylum Proteobacteria (e.g., *Enterobacteriaceae*), a major GUSB-producing bacterial group [46]. In addition, resolvin E1, a specialized pro-resolving mediator synthesized from n-3 PUFA, has recently been shown to significantly upregulate the expression of IAP, which is critical for the maintenance of bacterial homeostasis [54]. In this context, it is conceivable that CPT-11 treatment may cause a change in gut microbe composition to enhance GUSB activity by the increase in proinflammatory cytokines and the decrease in the expression and activity of IAP, while n-3PUFAs can exert opposing effects on inflammation, IAP activity and gut microbiota to decrease GUSB activity.

PICRUSt2 predicts the functional content of a metagenome from 16S rRNA amplicon sequencing data [55]. Functional orthologs are annotated in the KEGG Orthology database [56]. In line with lower GUSB-activity of cecal contents, PICRUSt2 predicted a decrease in the abundance of GUSB orthologs (ko1195) in response to CPT-11 after 11 days in the bacterial DNA extracted from cecal contents of mice with decreased n-6/n-3 ratio. Interestingly, we observed reduced fecal GUSB activity with the decreased n-6/n-3 ratio in response to six doses of CPT-11, but before diarrhea started. However, the fecal GUSB-activity was the same between groups at the baseline. These observations suggest that the decreased n-6/n-3 ratio is probably associated with a rapid decrease in bacterial metabolism in response to the introduction of CPT-11. This decrease appears to be sustained. GUSB activity levels were quite variable in humans but did correlate with CPT-11-related side effects for the patients with the highest GUSB activity level (high turnover microbiota metabotype) [35]. Therefore, stool GUSB activity levels in humans on omega-3 PUFA supplementation may represent a potential biomarker [35] to assess CPT-11 drug-related toxicity. However, this remains to be tested in a larger cohort. Likewise, future studies are needed in a tumor xenograft model to test whether the reduced n-6/n-3 ratio maintains the efficacy of CPT-11 while reducing the GUSB activity. However, it has been shown that GUSB inhibition by targeted bacterial GUSB inhibitors [46] prevents intestinal toxicity and maintains the antitumor efficacy of Irinotecan.

Intestinal mucosal barrier function plays a vital role in maintaining gut homeostasis. The active SN–38-induced epithelial damage leads to the invasion of enteric bacteria (e.g., *E. coli*) into the mucosa, recognition of pathogen-associated molecular patterns (PAMPs) by the toll-like receptors, activation NF-κB, and proinflammatory cytokine synthesis (e.g., TNF- α and IL-1β) [57,58]. TNF-α regulates chemotherapy-induced early intestinal endothelial damage responses, reduces epithelial oxygenation, and ultimately orchestrates the epithelial basal-cell death and injury process. All these events contribute to neutrophil recruitment to the site of injury, amplifying the damage and decrease in goblet cell number and function [7,59,60]. In our study, decreased goblet cell numbers and MUC2 gene expression in the WT+CPT-11 mice colon may indicate the discharge of mucins from the mucosal surface, which may contribute to the onset of diarrhea induced by CPT-11, and depletion of mucin stores may result in loss of integrity to the mucus barrier during mucositis. Moreover, lower expression of MUC2 and acid mucins and higher levels of markers of colonic inflammation in WT+CPT-11 mice are consistent with the fact that the expression of MUC2 can be inversely correlated with the severity of inflammation [57,61,62]. Likewise, tight junctions are mainly responsible for the restriction and modulation of intestinal permeability [63]. The disruption of the intestinal barrier function is characterized by increased intestinal permeability [63,64]. Numerous studies have intensively demonstrated mucosal inflammation [38,57,65], goblet cell dysfunction [7,66], altered tight junction protein expression, and increased intestinal permeability [63,64] during chemotherapy. This study showed that decreased n-6/n-3 ratio prevented mucosal inflammation, goblet cell dysfunction, decreased intestinal tight junction gene expression and the increase in intestinal permeability in irinotecan-treated mice. Previous studies sufficiently demonstrated the decreased n-6/n-3 ratio’s preventive effects against reducing the intestinal mucosal barrier function [18,19,48]. In the current study, we showed that a decreased n-6/n-3 PUFA ratio reduced the CPT-11-induced decrease in microbial diversity and increase in LPS-producing proteobacteria [10,67,68], which are associated with mucosal inflammation and impaired intestinal barrier function [18,19]. This study demonstrated the inhibitory effects of a decreased n-6/n-3 ratio (achieved by both transgenic and fish oil supplementation) on the GUSB-producing taxa [18,19] and associated GUSB activity with reduced GIT.

Moreover, the observed co-occurrences between GUSB taxa, antioccurrences between GUSB taxa and intestinal health maintaining taxa, and negative associations of tissue n-6/n-3 ratio with GUSB-producing taxa in the WT+CPT-11 were not observed in the network for FAT-1+CPT-11 group, suggesting that the decreased n-6/n-3 ratio favors balanced or healthy microbe–microbe and host–gut microbiome interactions as we have recently shown [19,68]. From a bench-to-bedside perspective, our discoveries have several important clinical implications. First, fecal GUSB activity and tissue n-6/n-3 ratio could be utilized as biomarkers to micro-type patients with cancer receiving Irinotecan and improving its pharmacokinetics or reducing its toxicity. Next, with n-3 PUFA supplementation or reduction in dietary n-6 PUFA, the reduction in the n-6/n-3 ratio and associated fecal GUSB activity could be achieved before starting the CPT-11 chemotherapy to decrease the risk of developing GIT. Several preclinical models reported therapeutic measures modulating GIT. Among these, GUSB was a central therapeutic focus to alleviate irinotecan-induced diarrhea to mitigate Irinotecan’s life-threatening GIT [35,69]. Indeed, recent data confirmed that the depletion of the intestinal GUSB activity by antibiotic treatment reduced this GIT [22,35,70]. However, broad-spectrum antibiotics administration can indiscriminately eliminate much enteric bacterial microbiota, open niches for pathogenic species such as *Clostridium difficile*, and negatively impact patients’ health [35]. Thus, it is essential to specifically target intestinal bacterial GUSB activity without developing significant dysbiosis that may enable the selection and translocation of pathogenic bacterial species [35]. Given that GUSB is found in most Enterobacteriaceae members, explicitly targeting Enterobacteriaceae and associated GUSB activity by altering tissue n-6/n-3 PUFA ratio seems an effective and safe option to prevent irinotecan-induced enteric toxicity. Moreover, reactivation of inactive form of several drug metabolites into toxic form is mediated by GUSB, so the decreased n-6/n-3 ratio-induced reduction in GUSB-producing bacteria and GUSB activity could be a novel strategy to prevent GIT not only with CPT-11 but also with commonly used chemotherapeutic drugs (e.g., mycophenolate mofetil) [70].

## 4. Conclusions

High-throughput 16S rRNA amplicon sequencing-based host–gut microbiome interaction (HMI) analyses of our unique FAT-1 transgenic animals treated with CPT-11 uncover a preventive effect of a decreased tissue omega-6/omega-3 PUFA ratio on CPT-11 induced GIT. A decreased tissue n-6/n-3 ratio leads to the prevention of CPT-11-induced increase in GUSB-producing taxa and GUSB activity and imbalances in HMIs. Subsequently, the reduction in intestinal mucosal barrier dysfunction and goblet cell dysfunction prevents the development of CPT-11-induced clinical manifestations (weight loss, late-onset bloody diarrhea, and death). Overall, this study demonstrates the importance of balancing the tissue omega-6/omega-3 PUFA ratio to maintain good intestinal health and manage chemotherapy-induced gut toxicities.

## 5. Materials and Methods

### 5.1. Animals

Transgenic FAT-1 mice were generated as described previously [25] and back-crossed onto a C57BL/6 background. Heterozygous FAT-1 mice were mated to obtain wild type (WT) and FAT-1 littermates. Mice have been housed in the Massachusetts General Hospital (MGH) animal facility in a biosafety level 2 room in hardtop cages. Mice were maintained at 22–24 °C in a 12-h light/dark cycle and allowed *ad libitum* access to food and water. Male mice were chosen for all experiments. Tissue n-3 PUFA levels in FAT-1 mice were confirmed using gas chromatography (GC). All animal procedures were carried out following the MGH Animal Committee guidelines and Institutional Animal Care and Use Committee (IACUC) approval.

### 5.2. Animal Experiments

#### 5.2.1. Studying the Role of Decreased Tissue n-6/n-3 PUFA Ratio in Preventing CPT-11-Induced Gut Toxicities

Irinotecan (CPT-11, MGH Pharmacy, Boston, MA, USA) was purchased as a hydrochloride salt (>99% HPLC purified grade). CPT-11 dissolved in double-distilled water (ddH20) to make a 20 mg/mL stock and stored at room temperature for a maximum of 2 h before use. Healthy, five-week-old wild type (WT) and FAT-1 transgenic (Tg) mice were arranged into four groups (*n* = 10 per group; 5mice per cage). Mice from WT+CPT-11 and FAT-1+CPT-11 groups were subjected to 9 consecutive days of treatment with once-daily intraperitoneal injections (i.p.,) of 50 mg/kg CPT-11. WT and FAT-1 control groups were treated with i.p., injections of vehicle (ddH_2_O) for CPT-11. The total injected volume was identical for each animal. As described previously by Wallace et al., a dosing scheme of 50 mg/kg/day, once daily for nine days, was chosen to accelerate diarrhea’s onset while preventing death [22] and 50 mg/kg CPT-11 in mice is roughly equivalent to the 5 mg/kg typical human CPT-11 dose based on differences in body surface area [22]. All four groups of mice were fed identical diets [AIN-76 diet (with 10% corn oil) (Harlan Laboratories, Indianapolis, IN] throughout the experiment. Drugs were administered between one and two hours after starting the light cycle to control CPT-11 chronotoxicity (circadian effects of Irinotecan) [46,71]. Mice were examined daily for signs of diarrhea (fecal staining of the skin, loose, watery stool) and bloody diarrhea (black sticky stool or frank blood). The percentage of each diarrhea severity level on day 11 was calculated as described previously [72]. All animals were closely monitored for signs of GIT (GI symptoms and changes in gross appearance) and moribund condition and were regularly weighed; animals were euthanized if they lost 20% body weight. Before terminal dissections, animals were deeply anesthetized using ketamine-xylazine, and cardiac puncture was used to collect blood. After removing the luminal contents, the large intestinal tissue was sectioned into the cecum and proximal and distal colon and stored in the 10% formalin for histopathological analysis. A portion of dissected intestinal tissues was flash-frozen in liquid nitrogen and then stored at −80 °C for tissue fatty acid and cytokine analysis. Another portion of dissected intestinal tissues stored in Trizol^®^ reagent was also flash-frozen in liquid nitrogen and then stored at −80 °C for mRNA analysis. Cecal contents collected at 11 days were flash-frozen in liquid nitrogen and then stored at −80 °C for gut microbiome analysis. The fecal samples collected at the baseline (before CPT-11) and 6 days (after CPT-11 and before diarrhea started) and cecal contents collected at 11 days (post CPT-11 and diarrheal events) were subjected to bacterial beta-glucuronidase (GUSB) activity measurements. The contents of the colon collected at 11 days were subjected to bacterial culture experiments as fecal samples were not available because of diarrheal events.

#### 5.2.2. Determination of CPT-11-Induced Lethality

In a separate experiment, 7-week-old male WT and FAT-1 mice (*n* = 5 per group; 3 or 2 mice per cage) were given CPT-11 (50 mg/kg) once per day for four constitutive days by i.p., injections. CPT-11–induced lethality was determined throughout 3 week. Kaplan-Meier survival curves and Log-rank (Mantel-Cox) test (GraphPad Prism 8) were used to graph the survival vs. time curves and calculate the *p*-value.

#### 5.2.3. Studying the Preventive Effects of Fish Oil Supplemented Diet against the CPT-11-Induced Gut Toxicities

Male, individually-housed five-week-old wild type (WT) mice (*n* = 5 per group) were fed either the omega-6 [10% corn oil (CO)] or omega-3 [5% CO+5% fish oil (FO)] PUFA supplemented diet either for 10 days in the first experiment or 2 months in a separate 2nd experiment. These mice were then subjected to nine consecutive days of treatment with 50 mg/kg CPT-11 (i.p., injections) to assess the preventive effects of decreased tissue n-6/n-3 ratio induced by FO supplemented diet against CPT-11-induced gut toxicities. The diets were prepared, as described previously [18]. Before terminal dissections, animals were deeply anesthetized using ketamine-xylazine, and cardiac puncture was used to collect blood. After gently removing the luminal contents, the large intestinal tissue was sectioned into the cecum and proximal and distal colon and stored in the 10% formalin for histopathological analysis. A portion of dissected intestinal tissues was flash-frozen in liquid nitrogen and then stored at −80 °C for tissue fatty acid and cytokine analysis. Another portion of dissected intestinal tissues stored in Trizol^®^ reagent was also flash-frozen in liquid nitrogen and then stored at −80 °C for mRNA analysis. Cecal contents collected at 11 days were flash-frozen in liquid nitrogen and then stored at −80 °C for gut microbiome analysis. The fecal samples collected at the baseline and 6 days and a portion of cecal contents collected at 11 days were subjected to bacterial beta-glucuronidase (GUSB) activity measurements. The contents of the colon collected at 11 days were subjected to bacterial culture experiments.

### 5.3. Histopathological Analysis

Formalin-fixed and paraffin-embedded cecum and proximal and distal colon samples were stained with hematoxylin and eosin (MGH Core, Boston, MA, USA) and examined by an independent, experienced pathologist (Dr. Bhan AK) blinded to group assignment to arrive at a histologic score as described previously [22]. For cell infiltration of inflammatory cells, rare inflammatory cells in the lamina propria were counted as 0; increased numbers of inflammatory cells, including neutrophils in the lamina propria as 1; a confluence of inflammatory cells, extending into the submucosa as 2; and a score of 3 was given for transmural extension of the inflammatory cell infiltrate. For epithelial damage, the absence of mucosal damage was counted as 0, discrete focal lymphoepithelial lesions were counted as 1, mucosal erosion/ulceration was counted as 2, and a score of 3 was given for extensive mucosal damage and extension through deeper structures of the bowel wall. The two subscores were added, and the combined histologic score ranged from 0 (no changes) to 6 (extensive cell infiltration and tissue damage). Inflammatory infiltrate cell count per field was calculated [59]. Paraffin sections of the proximal colon were stained with Alcian blue (AB) and periodic acid–Schiff (PAS) (MGH Core, Boston, MA, USA) for goblet cells, and Alcian blue-stained goblet cells were expressed per 10 villus-crypt units as described previously [73]. Both AB and Periodic Acid—Schiff (PAS) staining were used to identify any functional differences in goblet cells. AB stains blue for acid mucins. PAS stains purple for a mixture of neutral and acid mucins, and red/magenta color for neutral mucins alone [62]. The examination of tissues with both AB and PAS might indicate changes in the distribution or pattern of expression of neutral and acid mucins, which are indicative of certain pathological conditions [62,74]. The H&E-stained colonic tissue section was evaluated for crypt length changes, as described previously [75]. The paraffin sections of proximal colon tissues were stained with Gram stain (MGH Core, Boston, MA, USA) and evaluated by microscopy for the presence of Gram-negative (pink) or Gram-positive (blue) bacteria [76].

### 5.4. Extraction and Purification of DNA from Cecal Contents

Bacterial genomic DNA was extracted from cecal contents (~180 mg) using the QIAamp DNA Stool Mini Kit (Qiagen, Valencia, CA, USA), following the manufacturer’s instructions. To increase effectiveness, the lysis temperature was increased to 95 °C. The eluted DNA was treated with RNase, concentration was determined by absorbance at 260 nm (A260), and purity was estimated by determining the A260/A280 ratio with a Nanodrop spectrophotometer (Biotek, Winooski, VT, USA). The DNA samples with a 260/280 ratio close to 2 packed with dry ice were then shipped to APC Microbiome Ireland (University College Cork, Cork, Ireland) for 16S rRNA amplicon sequencing.

### 5.5. S rRNA Gene Amplicon Sequencing

The gDNA samples were subjected to16S rRNA amplicon sequencing, as previously mentioned [68]. Briefly, V3–V4 amplicons for Illumina sequencing were generated according to the 16S metagenomic sequencing library protocol (Illumina). An initial PCR reaction utilized primers specific for amplifying the V3–V4 region of the 16S rRNA gene (Forward primer 5′TCGTCGGCAGCGTCAGATGTGTATAAGAGACAGCCTACGGGNGGCWGCAG; reverse primer 5′GTCTCGTGGGCTCGGAGATGTGTATAAGAGACAGGACTACHVGGGTATCTAATCC). PCR product clean-up and purification were achieved using the Agencourt AMPure XP system (Labplan, Dublin, Ireland). A second PCR incorporated a unique indexing primer pair for each sample (Illumina Nextera XT indexing primers, Illumina, Sweden). The products were again purified using the Agencourt AMPure XP system. Samples were quantified using the Qubit broad range DNA quantification assay kit (Bio-Sciences, Dublin, Ireland). Following quantification, samples were pooled in equimolar amounts (10 nM) and sequenced at Clinical-Microbiomics, Copenhagen, Denmark, using Illumina MiSeq 2 × 300 bp paired-end sequencing. DNA extraction blanks, PCR blanks, and technical duplications for both extractions and PCRs were employed to ensure proper sample handling throughout the library preparation process. 

### 5.6. Bioinformatics

Three hundred-base pair paired-end reads were assembled using FLASH with parameters of a minimum overlap of 20 bp and a maximum overlap of 120 bp [77]. Sequences were processed using the Quantitative Insights Into Microbial Ecology version 2 (QIIME 2) pipeline (November 2020) [17,78]. Briefly, demultiplexed paired-end sequences were imported using Casava 1.8 format and denoised using DADA2 [79] to obtain an amplicon sequence variant (ASV) table. Singletons (ASV that were observed fewer than 2 times) and ASVs present in less than 10% of the samples were discarded. A phylogenetic tree was generated, and a naive Bayes taxonomy classifier pre-trained on the Silva 138 reference database (clustered at 99% similarity) was used to assign taxonomy to ASV [80]. An even sampling depth (sequences per sample) of 17,789 (FAT-1 mice studies) or 20,617 (FO studies) sequences per sample was used for assessing alpha- and beta-diversity measures. Pielou’s evenness index [81], Faith’s phylogenetic diversity (PD), observed features, and the Shannon diversity index was used to measure alpha-diversity. Kruskal–Wallis (all groups and pairwise) tests were used in QIIME2 to compare α-diversity indices. Beta-diversity was calculated using Bray–Curtis, Jaccard, and weighted and unweighted UniFrac metrics [82], and significant differences among groups were tested with a multivariate analysis of variance (PERMANOVA) in QIIME2. Three dimensional (3D) views of score plots were prepared using XLSTAT-3D Plot (Addinsoft Inc., New York, NY, USA) [83] and Emperor in QIIME2. Analysis of Composition of Microbiomes (ANCOM) [84] in QIIME2 was applied at different taxonomic levels to identify differentially abundant features (i.e., present in different abundances) across sample groups. DESeq2 R package with Benjamini–Hochberg false discovery rate (FDR) corrected *p* values < 0.05 [85] was also utilized to perform differential abundance analysis [86]. DESeq2 is more robust and powerful in identifying the differential features (i.e., lower false positives) [86]. Hierarchical clustering (HCN) was performed with the hclust function in R package stat [86]. 

Random Forest (RF) analysis is performed using the Random Forest R package [87] provided by MicrobiomeAnalyst [86]. RF is a supervised learning algorithm suitable for high-dimensional data analysis. It uses an ensemble of classification trees, each of which is grown by random feature selection from a bootstrap sample at each branch. Class prediction is based on the majority vote of the ensemble. RF also provides other useful information such as OOB (out-of-bag) error and variable importance measure. During tree construction, about one-third of the instances are left out of the bootstrap sample. This OOB data is then used as a test sample to obtain an unbiased estimate of the classification error (OOB error). Variable importance is evaluated by measuring the increase in the OOB error when it is permuted. The outlier measures are based on the proximities during tree construction. 

### 5.7. Microbial Functional Prediction with PICRUSt2

The PICRUSt2 software tool was used for predicting functional abundances using FASTA of ASVs and a BIOM table of the abundance of each ASV across each sample, obtained from QIIME2 processing of 16S rRNA gene sequencing data, as described previously [55,88]. A single script, called picrust2_pipeline.py, was able to run each of the 4 key steps such as (1) sequence placement, (2) hidden-state prediction of genomes, (3) metagenome prediction, and (4) pathway-level predictions. The prediction of KO relative abundances was performed with hidden-state prediction [89] and was used to infer pathway abundances [90]. The nearest-sequenced taxon index (NSTI) was calculated for each input ASV, and any ASVs with NSTI > 2 were excluded from the output by default. 

### 5.8. Multivariate Receiver Operator Characteristic (ROC) Curve-Based Biomarker Analyses

The ROC curve analyses were performed for automated important feature identification, and performance evaluation using MetaboAnalystR and random forest classification and PLS-DA feature ranking method [91]. The relative abundance data was used without normalization and data transformation and scaling. The ROC curves were generated by Monte-Carlo cross-validation (MCCV) using balanced sub-sampling and visualized using Prism 8.0 (GraphPad Software, Inc., San Diego, CA, USA). In each MCCV, two-thirds (2/3) of the samples are used to evaluate the feature importance. The top 2, 3, 5, 10, and 100 (max) essential features are then used to build classification models validated on the 1/3 of the samples left out. The procedure was repeated multiple times to calculate the performance and confidence interval of each model. The ROC curves were created based on the cross-validation (CV) performance and the 95% confidence interval computed for the model. The most accurate biomarker model with the highest AUC was selected, and the crucial features were ranked by mean importance measure.

### 5.9. RV Coefficient

The RV coefficient (XLSTAT version 2019.1, Addinsoft Inc., NY, USA) was calculated between the microbial genera (FDR-corrected *p*-value < 0.05) and the host parameters (markers of ME, LGCI, and MS). The RV coefficient is a multivariate generalization of the Pearson correlation coefficient [92].

### 5.10. Cox Proportional-Hazards Analysis

Univariate models of Cox Proportional-Hazards analysis [93] (XLSTAT version 2019.1) were performed to assess the association of the tissue n-6/n-3 PUFA ratio and baseline fecal GUSB levels with the risk for diarrhea episodes. A *p* value of less than 0.05 (Pr > Chi^2^) was considered statistically significant.

### 5.11. Co-Occurrence Network Inference

Microbiota community structure was evaluated by building co-occurrence networks of the most abundant genera (>0.5% mean relative abundance in the global dataset) using the Sparse Correlations for Compositional data (SparCC) algorithm [27]. SparCC assumes a sparse network (i.e., that many taxa are not correlated with one another) and uses a log-ratio transformation and performs iterations to identify taxa pairs that are outliers to background correlations. Pseudo-*p*-values were calculated using a bootstrap procedure with 999 random permutations and 999 iterations for each SparCC calculation. Significant associations were defined as positive SparCC correlations with a *p*-value < 0.05. An undirected network, weighted by SparCC correlation magnitude and heat-map was generated using R version 3.6.3 provided by MicrobiomeAnalyst [86].

### 5.12. β-Glucuronidase (GUSB) Activity Assay

The supernatant of a homogenized stool or cecal contents suspension was used for the GUSB assay. Briefly, a small amount of frozen content (mg) was measured, and then the ice-cold GUSB assay buffer at a defined ratio was added before the sample was thawed (50 μL of dilution buffer was added to 1 mg of the sample). The sample was vigorously vortexed to prepare a homogenized suspension, which was then centrifuged at 10,000× *g* for 20 min at 4 °C, and the supernatant was collected and assayed for GUSB concentration using a fluorometric β-Glucuronidase activity assay kit (ab234625; Abcam, Cambridge, MA, USA) by following the manufactures’ instructions. The provided substrate, which is specific to GUSB, is cleaved into a fluorescent product in the presence of GUSB. Ten μL of the substrate working stock was added to the positive control, standards, and test samples, and then the fluorescence (excitation wavelength 330 nm and emission wavelength 450 nm; Perkin-Elmer, Waltham, MA, USA) was measured immediately after the addition of substrate for 0–60 min at 37 °C. All standards, controls, and samples were assayed in duplicate. One unit is the amount of GUSB that can cleave 1 μmol of substrate/min under the assay conditions at 37 °C.

### 5.13. Host–Microbiome Interaction Analysis 

The complex host–microbe interactions can be extricated by network-based analytical approaches, as described previously [68,83,94]. The host–microbiota interaction bipartite network was built using Spearman’s nonparametric rank correlation coefficient (*p* < 0.05) performed between host and genus-level microbial parameters for the comparisons between WT+CPT-11 and FAT-1+CPT-1 and CO+CPT-11 and FO+CPT-11. The nodes (filled squares) in the network represent host parameters (cyan or filled squares colored black) and microbes (olive or black and different shapes indicate different phylum). The lines (edges) represent statistically significant correlations (*p* < 0.05) and are colored blue for positive and red for negative correlations. The edge size reflects the magnitude of the correlation. The resulting network of statistically significant interactions was visualized in Cytoscape 3.8.2.

As described previously [68], the multiple factor analysis (MFA) was also utilized to assess the associations between the microbiome and host parameters. MFA using Spearman type principal component analysis was performed to superimpose the entire genus-level microbiome and host data. In the first part of the analysis, a Spearman type principal component analysis (PCA) was successively carried out for each dataset, which stores the value of the first eigenvalue of each analysis to then weigh the various datasets in the second part of the analysis, where the weighted PCA on the columns of all the datasets leads to each indicator variable having a weight that is a function of the frequency of the corresponding category. The coordinates of the projected points in the space resulting from the multiple factor analysis are displayed. The projected points correspond to projections of the spaces’ observations reduced to the dimensions of each dataset. Based on the weighted PCA’s eigenvalues, the first two factors (F1/F2) covered this analysis’s variability. To test whether the two groups with the superimposed microbiome and host data were separated from each other, Mann–Whitney testing was performed on the coordinates of all the observations’ projected points, and *p* values were obtained using 10,000 Monte Carlo simulations. One end of each line for an observation indicates the host data (differently colored to indicate the groups), and another end (dark yellow) indicates the microbiota. All analyses were accomplished using the XLSTAT-R software version 2019.1 (Addinsoft Inc., New York, NY, USA) [83].

### 5.14. Relative Quantification of Bacterial Species Carrying GUSB Genes in Cecal Microbiota

Relative quantification of GUSB-producing bacteria was performed on cecal genomic DNA as previously described [83,95]. Briefly, qRT-PCR was performed with a PRISM 9000 Light Cycler (Applied Biosystems, USA) using the iTaq Universal SYBR Green Supermix (Bio-Rad, Hercules, CA, USA) and degenerate primers explicitly targeting the bacterial GUSB gene (forward: 5-TATTTAAAAGGITTYGGIMRICAYGAR-3, reverse: 5-CCTTCTGTTGTIKBRAARTCIGCRAAR-3) [41,95] and total bacteria (forward: ACTCCTACGGGAGGCAGCAGT, reverse: ATTACCGCGGCTGCTGGC). Samples and controls were run in duplicate in total reaction volumes of 20 μL/well, containing 500 nM primers and 40 ng genomic DNA. PCR amplification was performed using the following conditions: initial denaturation (3 min at 94 °C), then 35 cycles of denaturation (30 s at 94 °C), ramped annealing (20 s at 55 °C, 5 s at 50 °C, and 5 s at 45 °C), and elongation (1 min at 72 °C) and a final extension (7 min at 72 °C). By subtracting the cyclic threshold (Ct) values of total bacteria from the Ct values of GUSB-gene, we estimated and compared the Δ-Ct values for relative quantification of the GUSB gene carrying bacterial group. For example, a higher Δ-Ct value indicates a lower abundance of GUSB gene-carrying bacteria. Melting-point curve analysis confirmed the specificity of the amplification products.

### 5.15. Determination of Cytokine Levels in the Colon

Cytokine levels were measured, according to Elian et al. [96]. This is briefly described as follows. The colonic tissue was homogenized with PBS (0.4 M NaCl, 10 mM Na_2_HPO_4_), containing anti-proteases (0.1 mM PMSF, 0.1 mM benzethonium chloride, 10 mM EDTA and Kallikrein Inhibitor Units aprotinin A) and 0.05% Tween 20. The samples were then centrifuged for 10 min at 12,000× *g*. The supernatant was analyzed for levels of TNF-α (LLOD:1.4 pg mL^−1^), IL-1β (LLOD: 9.4 pg mL^−1^), IL-6 (LLOD: 0.2 pg mL^−1^), MCP-1 (LLOD: 3.7 pg mL^−1^), and IL-10 (LLOD: 1 pg mL^−1^) by Bio-Plex immunoassays (assay range: 2–3000 pg/mL; intra-assay C.V.: <10%; inter-assay C.V.: <30%) formatted on magnetic beads (Bio-Rad Laboratories Inc., Hercules, CA, USA), following the manufacturer’s instructions [84]. Xponent software (Luminex, Austin, TX, USA) was used for data acquisition and analysis. For all the assays mentioned previously, 5–6 standards, including blank (negative control), were used.

### 5.16. Determination of the Fatty Acid Composition of Mouse Tissues and Diets

Fatty acid profiles of mouse diets and tail and colon tissues were analyzed by gas chromatography (GC), as described previously [18,97]. Briefly, tissue or food samples were ground to powder under liquid nitrogen and subjected to total lipid extraction and fatty acid methylation by 14% boron trifluoride (BF3)-methanol reagent (Sigma-Aldrich) at 100 °C for 1 h. Fatty acid methyl esters were analyzed using a fully automated HP5890 gas chromatography system equipped with a flame-ionization detector (Agilent Technologies, Palo Alto, CA, USA). The fatty acid peaks were identified by comparing their relative retention times with the mixed commercial standards (NuChek Prep, Elysian, MN, USA), and the area percentage for all resolved peaks were analyzed by using a PerkinElmer M1 integrator. The fatty acids examined as total n-6 PUFA with GC include: Linoleic acid (C18:2n6), Gamma-linolenic acid (C18:3n6), Eicosadienoic acid (C20:2n6), Dihomo-gamma-linolenic acid (C20:3n6), Arachidonic acid (C20:4n6), Docosadienoic acid (C22:2n6), Adrenic acid (C22:4n6) and Docosapentaenoic acid (22:5n6). The fatty acids examined as total n-3 PUFA with GC include: α-Linolenic acid (C18:3n3), Eicosatrienoic acid (C20:3n3), Eicosapentaenoic acid (C20:5n3), Docosapentaenoic acid (C22:5n3), and Docosahexaenoic acid (C22:6n3).

### 5.17. Gene Expression Analysis by RT-qPCR

Total RNA was isolated from colon tissue samples (~50 mg) stored in Trizol^®^ at −80°C and using TRIzol reagent (Invitrogen Life Technologies, Grand Island, NY, USA), following the manufacturer’s instructions. For RNA extraction, tissues were thawed in Trizol and homogenized using Kontes Pellet Pestle (Fisher Scientific, Hampton, NH, USA). RNA concentrations and purity were estimated by determining the A260/A280 ratio with a Nanodrop spectrophotometer (Biotek). Reverse transcription of RNA samples with a 260/280 ratio close to 2 was performed using the iScript cDNA Synthesis Kit (Bio-Rad). Real-Time-qPCR was carried out using SYBR Green in a PRISM 9000 Light Cycler (Applied Biosystems), following the manufacturer’s instructions. Primers for ZO-1 (forward: TCTACGAGGGACTGTGGATG, reverse: TCAGATTCAGCAAGGAGTCG) [18], Occludin (forward: ACCCGAAGAAAGATGGATCG, reverse: CATAGTCAGATGGGGGTGGA), claudin-1 (forward: TCTACGAGGGACTGTGGATG, reverse: TCAGATTCAGCAAGGAGTCG), MUC2 (forward: CTGACCACACCTTCCTGGAC, reverse: GATCAGCCGGATCTGCCTAC) [98], MUC4 (forward: GAGGGCTACTGTCACAATGGAGGC, reverse: GAGGGCTACTGTCACAATGGAGGC) [98], TLR4 (forward: CGCTTTCACCTCTGCCTTCACTACAG, reverse: ACACTACCACAATAACCTTCCGGCTC) [18], NF-kB p65 (forward: ATGCGCTTCCGCTACAAGTG, reverse: ACAATGGCCACTTGTCGGTG) [99] were synthesized by Invitrogen, USA. All samples were processed in triplicate and normalized to β -actin (forward: GAGAAGATCTGGCACCACACC, reverse: GCATACAGGGACAGCACAGC) [84]. The 2^(−ΔΔCT)^ method was used to determine the relative mRNA levels.

### 5.18. Measurement of LPS Concentration

Serum LPS concentrations were measured with a ToxinSensor Chromogenic Limulus Amebocyte Lysate (LAL) Endotoxin Assay Kit (GenScript), following the manufacturer’s instructions [83]. Briefly, to minimize inhibition or enhancement by contaminating proteins, the samples were diluted 10- to 50-fold with endotoxin-free water, adjusted to the recommended pH, and heated for 10 min at 70 °C. The lyophilized endotoxin standard was dissolved by adding 2 mL of LAL reagent water and mixed thoroughly for 15 min with a vortexer to obtain an endotoxin stock solution. LAL reagents were added to the serum and incubated at 37 °C for 45 min, and the absorbance was read at 545 nm. A spiked control at 0.45 EU/mL was included for each sample to check that no significant inhibition or activation occurred. The lower limit of detection (LLOD) was 0.01 EU mL^−1^. The coefficient of variation (C.V.) equals 100 times the standard deviation of a group of values divided by the mean and is expressed as a percent. The C.V. absorbance was less than 10%.

### 5.19. Immunohistochemistry (IHC) 

IHC analysis was performed as described previously [19]. Paraffin blocks obtained from formalin-fixed colonic tissues (*n* = 3 per control groups; *n* = 4 per CPT-11 treated groups) and primary antibodies for beta-glucuronidase (GUSB; 1:100; sc-374629), from Santa Cruz Biotechnology, Inc. (Dallas, TX, USA) and for Myeloperoxidase [30] (MPO; 1:100; GTX75318) from GeneTex (San Antonio, TX, USA) were given to Massachusetts General Hospital (MGH) Core, Boston, MA, USA. Briefly, deparaffinize slides, citrate antigen retrieval solution in a pressure cooker, dual endogenous enzyme Block (5 min), normal goat serum (20 min), incubation with primary antibody overnight at 4 °C and then rabbit polymer HRP (30 min) followed by DAB+ (5–10 min). All samples were photographed using an immunofluorescence microscope (LSM710; Zeiss) and analyzed for GUSB, and MPO with the help of a pathologist fellow at MGH Core.

### 5.20. Bacterial Culture

Bacterial culture was performed as described previously [18]. Individual colonic content samples were collected directly in sterile microfuge tubes containing 200 μL of Brain Heart Infusion (BHI) media and kept on ice. The weight of each sample was determined. Additional BHI media was added to each tube, obtaining a specific weight/volume ratio (100 μL BHI per 10 mg sample). Samples were vortexed, serially diluted to 100-fold, plated on MacConkey agar (BD, NJ, USA), then incubated in ambient air for 24 h at 37 °C to enumerate *Enterobacteriaceae*. Bacterial colonies were counted using Digital Bacterial Colony Counter and expressed as colony-forming units (CFU)/Gram sample.

### 5.21. Statistics and Reproducibility

Data were expressed as mean ± standard error of the mean (SEM). A Violin plot with lines at the median (dashed lines) and quartiles (complete lines) was also used to express the data. The two groups’ differences were always analyzed using the nonparametric Mann–Whitney test or regular two-way analysis of variance (ANOVA). Statistical differences among more than two groups were evaluated by ordinary or repeated measures of one-way or two-way ANOVA (with Geisser-Greenhouse correction to assume no equal variability in differences) and nonparametric Kruskal–Wallis tests. When we performed statistical analysis on data obtained from FAT-1 mice experiments with 4 groups, each significant result obtained between any 2 groups with tests mentioned previously was also subjected to an unequal variance (Welch) *t*-test. We decided to always use the Welch *t*-test as part of the experimental planning as we used only 3 mice per group for control WT and FAT-1 groups. The Welch *t*-test assumes that both data groups are sampled from Gaussian populations but does not assume those two populations have the same standard deviation. When the Welch *t*-test showed a *p*-value < 0.05 or the difference between means was out of the 95% confidence interval range, the results were considered non-significant and ignored. The ImageJ software package was utilized to draw scale bars on histopathological pictures and calculate staining intensity scores from the IHC analysis. Univariate analyses (*t*-tests, ANOVA, and correlation analysis) were performed using Prism 9.0 (GraphPad Software, Inc., San Diego, CA, USA). The significance was considered to be at *p* < 0.05. 

## Figures and Tables

**Figure 1 ijms-23-05332-f001:**
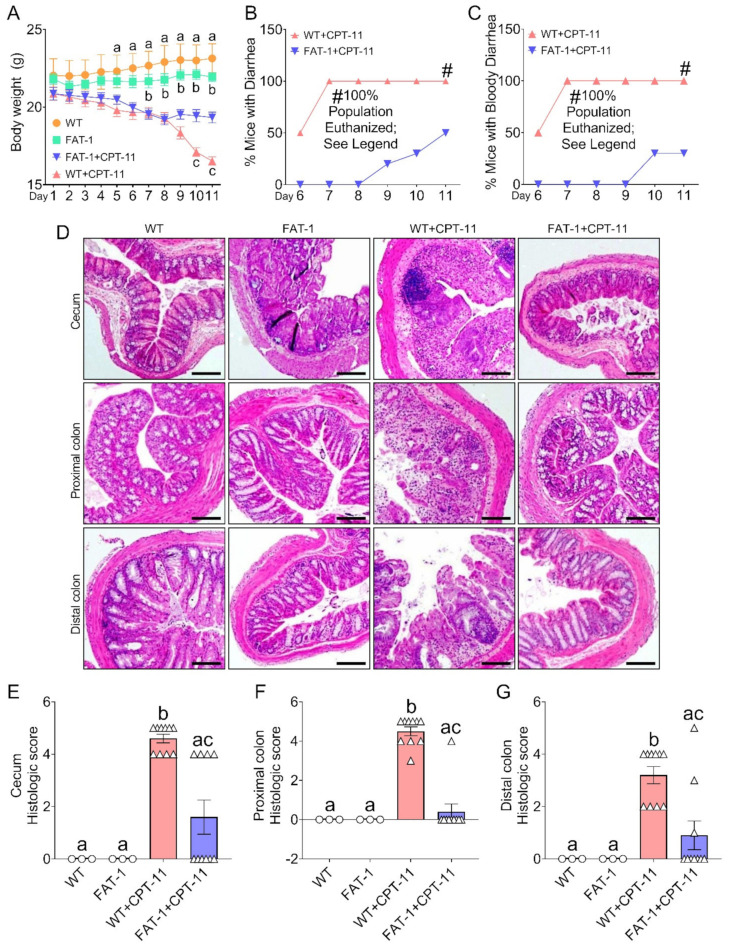
**Decreased tissue n-6/n-3 ratio reduces CPT-11-induced gastrointestinal toxicities**. Male, five-week-old wild type (WT) and FAT-1 transgenic mice fed an identical corn oil diet were arranged into four groups. Mice from WT+CPT-11 and FAT-1+CPT-11 groups and WT and FAT-1 control groups were subjected to nine consecutive days of treatment with intraperitoneal injections (i.p.,) of 50 mg/kg CPT-11 and vehicle for CPT-11, respectively. Please refer to Appendix A for experimental design. (**A**) Body weight was monitored every day. (**B**,**C**) CPT-11 produced diarrhea and bloody diarrhea starting after six days and peaking at seven days in the WT+CPT-11 group. In contrast, the incidence of diarrhea and bloody diarrhea were reduced in the FAT-1+CPT-11 group. The vehicle alone caused no bloody diarrhea in the control of WT and FAT-1 groups. By days 8 to 11, mice in the WT+CPT-11 group began to suffer from severe lethargy and lack of movement; by day 11, all mice in that group were euthanized. (**D**) Intestinal tissue was sectioned into the cecum and proximal and distal colon, and the morphology of the intestines was photographed after H&E staining. FAT-1+CPT-11 group showed a healthy glandular structure, but the WT+CPT-11 group showed highly disrupted tissues. (**E**–**G**) Histopathological scores for cecum and proximal and distal colonic inflammatory infiltration and epithelial damage. Data are shown as mean ± standard error of the mean. Panel (**A**) data were analyzed by repeated-measures two-way ANOVA followed by Tukey’s multiple comparisons test and *p* < 0.05 for a (WT vs. WT+CPT-11), b (FAT-1 vs. FAT-1+CPT-11), and c (WT+CPT-11 vs. FAT-1+CPT-1). Panel (**B**,**C**) data were analyzed by regular two-way ANOVA followed by Sidak’s multiple comparisons test and *p* < 0.05 at all-time points. Data with different superscript letters are significantly different (*p* < 0.05) according to ordinary two-way (**E**–**G**) followed by Tukey’s multiple comparisons test. Scale bar for images: 1500 μm.

**Figure 2 ijms-23-05332-f002:**
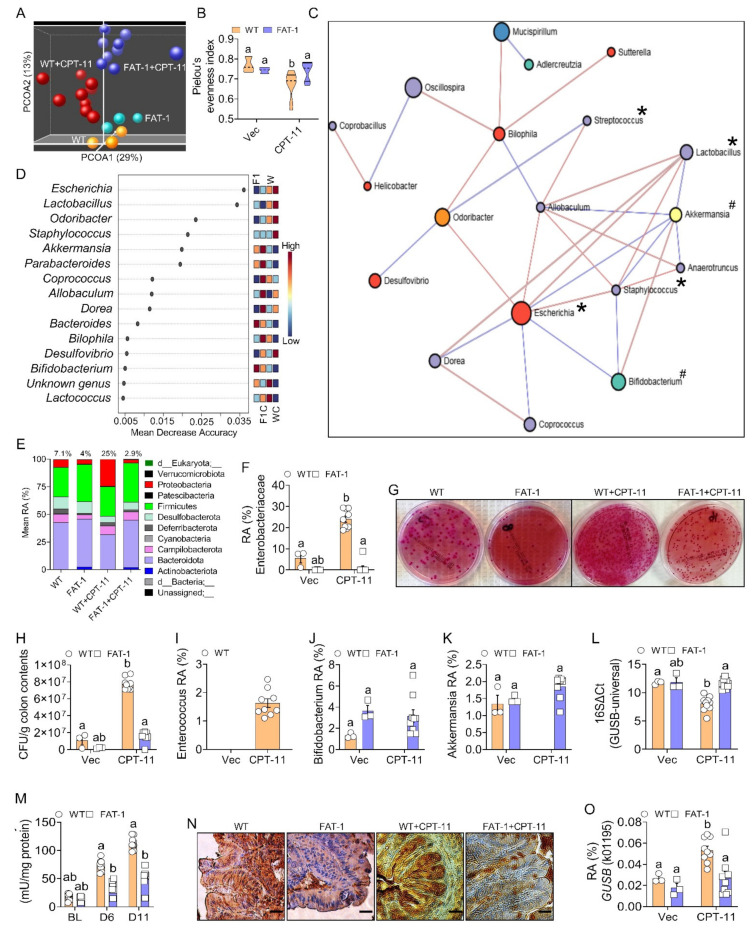
**Decreased tissue n-6/n-3 ratio prevents CPT-11-induced alterations in the gut microbiome**. (**A**) Principal coordinates analysis (PCOA) plot showing the results of Bray–Curtis distance-based analysis of beta diversity metrics. (**B**) Violin plot with lines at the median (dashed lines) and quartiles (complete lines) showing the differences in the Pielou’s evenness index. (**C**) Microbe–microbe interactions network [SparCC correlation analysis (WT+CPT-11 vs. FAT-1+CPT-11)]. Each node (*, GUSB-producing taxa; #, healthy gut making taxa) represents a taxon (colored based on the phylum level and sized based on the number of connections to that taxon). Two taxa are connected by an edge (co-occurrences: red; anti-occurrences: blue; *p*-value < 0.05 and correlation threshold 0.3; size reflects the magnitude). (**D**) Random Forests classification of taxa (genus level) in the vehicle (W and F1) or CPT-11 (WC and F1C) treated groups. (**E**) Phyla detected in the control and CPT-11 treated WT and FAT-1 mice. The numbers above each group show the relative abundance (RA) of the Proteobacteria phylum. (**F**–**K**) RA of differentially abundant (ANCOM test by QIIME2) bacterial groups such as Enterobacteriaceae (**F**) with representative colonic luminal contents MacConkey agar culture plate photos (**G**) showing the difference (**H**) in the growth of *Escherichia Coli* (pink colonies), *Enterococcus* (**I**), *Bifidobacterium* (**J**) and *Akkermansia* (**K**). (**L**) RA of beta-glucuronidase (GUSB)-producing bacteria measured using qPCR. (**M**) The difference in GUSB activity was measured at baseline (BL) and days (d) 6 using stool samples and at days 11 using cecal contents. (**N**) Immunohistochemical staining-based GUSB gene expression patterns in the proximal colon. (**O**) RA of GUSB (K01195) gene predicted using PICRUSt2. Data are shown as mean ± standard error of the mean. Data with different superscript letters are significantly different (*p* < 0.05) according to the Kruskal–Wallis test (**B**) or Mann–Whitney test, or ordinary two-way (**M**) ANOVA followed by Sidak’s multiple comparisons test. Scale bar for images in (**J**) panel: 2000 μm.

**Figure 3 ijms-23-05332-f003:**
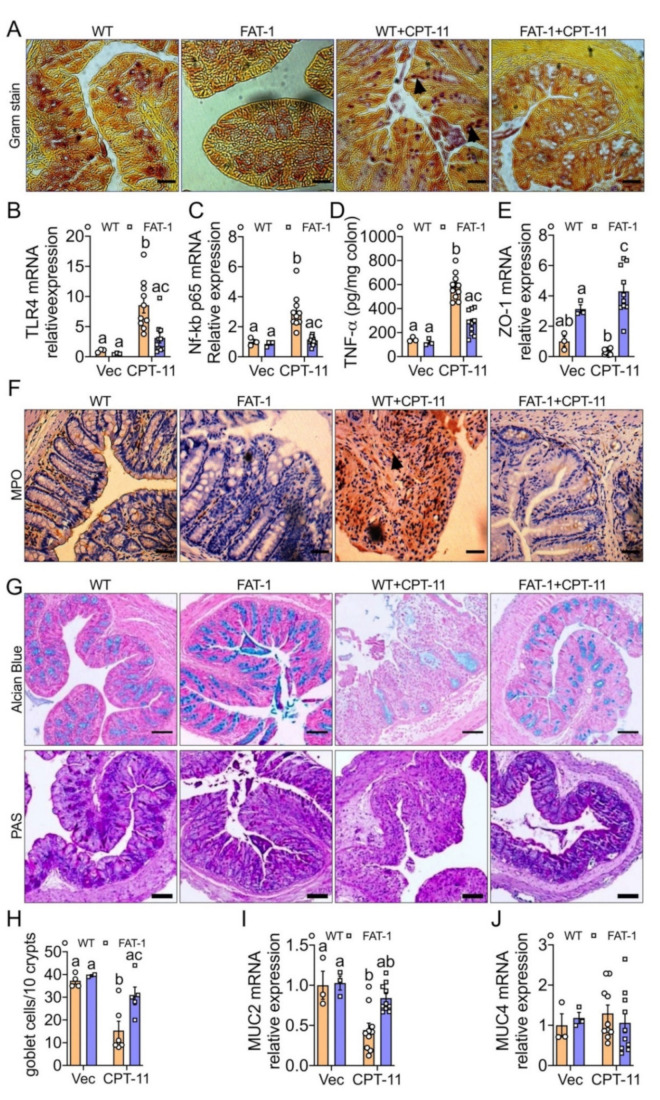
**Decreased n-6/n-3 ratio prevents CPT-11-induced intestinal mucosal barrier dysfunction**. (**A**) Gram staining was performed on paraffin sections of proximal colonic tissue sections and evaluated by microscopy for the presence of Gram-negative (pink) bacteria (arrowheads). (**B**–**F**) The markers of colonic mucosal inflammation [toll-like receptor 4 (TLR 4), NF-kB-p65 and TNF-α], barrier integrity [Zonulin 1 (ZO-1] and inflammatory cells infiltration were analyzed by real-time polymerase chain reaction (PCR), Bio-Plex immunoassays, and Myeloperoxidase (MPO) immunohistochemistry assays to quantify the levels of messenger RNA (mRNA) (**B**,**C**,**E**), cytokine (**D**) and infiltrated neutrophils (arrowhead) (**F**), respectively. (**G**,**H**) Representative pictures showed the differences in colonic Alcian Blue (acid mucins by blue color) and PAS (mixture of acid and neutral mucins by purple color) stained goblet cells and scattered dot-plot showing the numbers of Alcian blue-stained goblet cell counts. (**I**,**J**) Differences in the Mucin2 (MUC2) and Mucin4 (MUC4) relative gene expression. Data are shown as mean ± standard error of the mean. According to ordinary two-way ANOVA, data with different superscript letters are significantly different (*p* < 0.05), followed by Tukey’s multiple comparisons tests. The scale bar for images in (**A**,**F**,**G**) panels is 3000 μm and 1000 μm.

**Figure 4 ijms-23-05332-f004:**
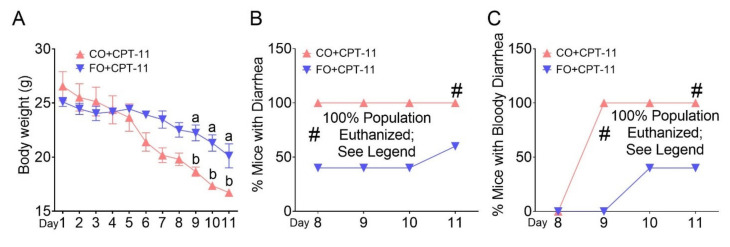
**Dietary n-3 PUFA supplementation reduces CPT-11-induced gastrointestinal toxicities**. Male, five-week-old wild type (WT) mice were divided into two groups (*n* = 5 per group) and were fed omega-6 [10% corn oil (CO)] and omega-3 [5% CO+5% fish oil (FO)] PUFA supplemented diets for two months. These mice were then subjected to nine consecutive days of treatment with intraperitoneal injections (i.p.,) 50 mg/kg CPT-11. (**A**) Bodyweight was monitored every day. (**B**) CPT-11 produced diarrhea starting as well as peaking after eight days in the CO+CPT-11 group. (**C**) CPT-11 produced bloody diarrhea starting after eight days and peaking at nine days in the CO+CPT-11 group, whereas the incidence of diarrhea and bloody diarrhea were reduced in the FO+CPT-11 group. By days 9 to 11, mice in the CO+CPT-11 group began to suffer from severe lethargy and lack of movement; by day 11, all mice in that group were euthanized. (**D**) Large intestinal tissue was sectioned into the ice, cecum, proximal and distal colon, and the intestines’ morphology was photographed after H&E staining. FO+CPT-11 group showed a healthy glandular structure, but the CO+CPT-11 group showed highly disrupted tissues. (**E**–**G**) Histopathological scores for cecum and proximal and distal colonic inflammatory infiltration and epithelial damage. Data are shown as mean ± standard error of the mean. Data with different superscript letters are significantly different (*p* < 0.05) according to repeated measures, two-way ANOVA followed by Sidak’s multiple comparisons test (**A**). Regular two-way ANOVA followed by Sidak’s multiple comparisons test, *p* < 0.05 at all-time points (**B**,**C**). Nonparametric Mann–Whitney test (**E**–**G**). * *p* < 0.05, ** *p* < 0.01, Scale bar for images: 1500 μm.

**Figure 5 ijms-23-05332-f005:**
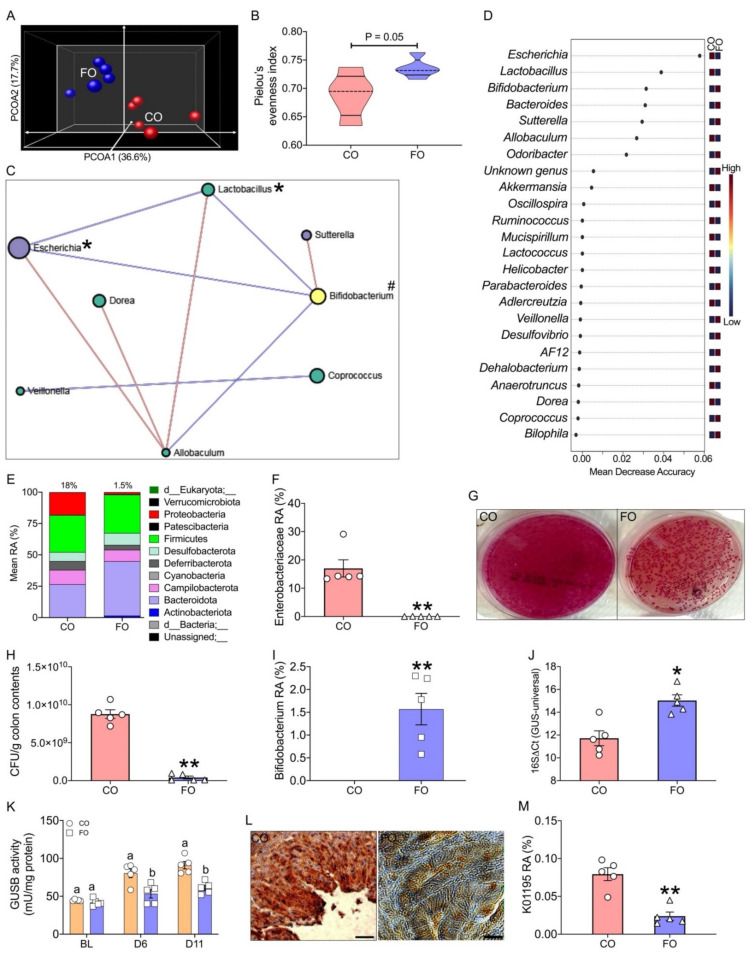
**Dietary n-3 PUFA supplementation reduces CPT-11-induced alterations in the gut microbiome**. (**A**) Principal coordinates analysis (PCOA) plot showing the results of Bray–Curtis distance-based analysis of beta diversity metrics. (**B**) Violin plot with lines at the median (dashed lines) and quartiles (complete lines) showing the differences in the Pielou’s evenness index (α-diversity). (**C**) Microbe–microbe interactions network [SparCC correlation analysis (CO+CPT-11 vs. FO+CPT-11)]. Each node (*, GUSB-producing taxa; #, healthy gut making taxa) represents a taxon (colored based on the phylum level and sized based on the number of connections to that taxon). Two taxa are connected by an edge (co-occurrences: red; anti-occurrences: blue; *p*-value < 0.05 and correlation threshold 0.3; size reflects the magnitude). (**D**) Random Forests classification of taxa (genus level) of CPT-11 treated CO and FO groups. (**E**) Composition summary showing the phyla detected in the CPT-11 treated CO and FO groups. The numbers above each bar show the relative abundance (RA) of the Proteobacteria phylum. (**F**–**I**) RA of differentially abundant bacterial groups such as Enterobacteriaceae (**F**) with representative colonic luminal contents MacConkey agar culture plate photos (**G**) showing the difference (**H**) in the growth of *Escherichia Coli* and *Bifidobacterium* (**I**), which is not detectable in CO group. (**J**) qPCR results showing the RA of beta-glucuronidase (GUSB)-producing bacteria. (**K**) The difference in GUSB activity was measured at baseline (BL) and days (d) 6 using stool samples and at days 11 using cecal contents. (**L**) Representative pictures are showing GUSB expression measured at proximal colon using the immunohistochemical technique. (**M**) RA of GUSB (K01195) gene predicted using PICRUSt2. Data are shown as mean ± standard error of the mean. Data with different superscript letters are significantly different (*p* < 0.05) according to the nonparametric Mann–Whitney test (* *p* < 0.05, ** *p* < 0.01) or ordinary two-way (**K**) ANOVA followed by Sidak’s multiple comparisons tests. Scale bar for images in (**J**) panel: 2000 μm.

**Figure 6 ijms-23-05332-f006:**
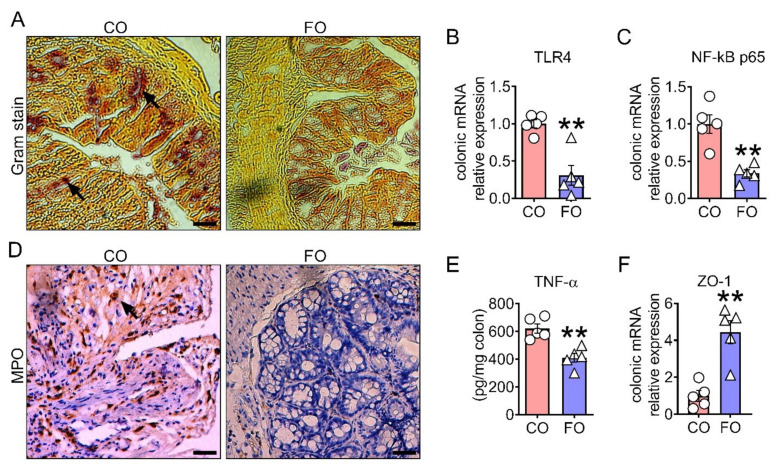
**Dietary n-3 PUFA supplementation reduces CPT-11-induced intestinal mucosal barrier dysfunction**. (**A**) Gram staining was performed on proximal colonic tissue sections and evaluated by microscopy for the presence of Gram-negative (pink) bacteria (arrowheads). (**B**–**F**) The markers of colonic mucosal inflammation [toll-like receptor 4 (TLR 4), NF-kB-p65 and TNF-α], barrier integrity [Zonulin 1 (ZO-1] and infiltration of inflammatory cells were analyzed by real-time polymerase chain reaction (PCR), Bio-Plex immunoassays and Myeloperoxidase (MPO) immunohistochemistry assays to quantify the levels of messenger RNA (mRNA) (**B**,**C**,**E**), cytokine (**D**) and infiltrated neutrophils (**E**), respectively. (**G**,**H**) Representative pictures showed the differences in colonic Alcian Blue (acid mucins by blue color) and PAS (mixture of acid and neutral mucins by purple color) stained goblet cells and scattered dot-plot showing the numbers of Alcian blue-stained goblet cell counts. (**I**,**J**) Differences in the Mucin2 (MUC2) and Mucin4 (MUC4) relative gene expression. Data are shown as mean ± standard error of the mean. ** *p* < 0.01, *** *p* < 0.001, Student’s *t*-test or nonparametric Mann–Whitney test. The scale bar for images in (**A**,**D**,**G**) panels are 3000 μm and 1000 μm, respectively.

**Figure 7 ijms-23-05332-f007:**
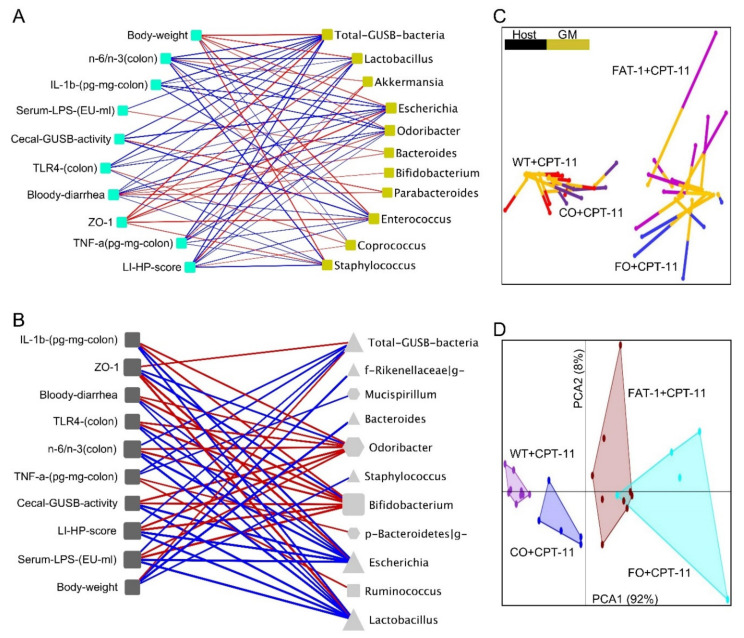
**Decreased tissue n-6/n-3 ratio reduces CPT-11-induced imbalances in the host–gut microbiome interactions**. (**A**,**B**) Host–microbiota interaction network built from Spearman’s nonparametric rank correlation coefficient (*p* < 0.05) between host parameters and entire microbial parameters (genus-level) of WT+CPT-11 vs. FAT-1+CPT-1 (**A**) and CO+CPT-11 vs. FO+CPT-11 (**B**) comparisons. Nodes (filled squares) in panel A represent host parameters (cyan) and microbes (olive). Nodes in panel B represent host parameters (filled squares colored black) and microbes (different shapes indicate different phylum and colored light black). Lines (edges) represent statistically significant correlations (*p* < 0.05) and are colored blue for positive and red for negative correlations. Edge size reflects the magnitude of the correlation. (**C**) Multiple Factor Analysis superimposing the host and gut microbiome (genus-level) data associated with a high tissue n-6/n-3 ratio (WT+CPT-11/CO+CPT-11 samples) and a low tissue n-6/n-3 PUFA ratio (FAT-1+CPT-11/FO+CPT-11 samples). Each line connects the host and microbial data from one sample. One end of each connecting line for an observation indicates the host (differently colored to indicate the groups), and another end (dark yellow) indicates the gut microbiota (GM). (**D**) Principal component analysis (PCA) of the host and gut microbiome (genus-level) data associated with a high n-6/n-3 ratio and a low n-6/n-3 PUFA ratio. (**E**,**F**) Biomarker analysis using multivariate [Random Forests (RF) classification with PLS-DA feature ranking method] receiver operator characteristic curve (ROC) based exploratory analysis performed on the combined host and microbial parameters. (**G**) Post-CPT-11 cecal contents beta-glucuronidase (GUSB) activity measurements were associated with a high n-6/n-3 ratio and a low n-6/n-3 PUFA ratio. (**H**) ROC curve generated with classical univariate ROC curve analysis showing the sensitivity and specificity for cecal contents GUSB activity. Data are shown as mean ± standard error of the mean. *** *p* < 0.001, nonparametric Mann–Whitney test.

**Figure 8 ijms-23-05332-f008:**
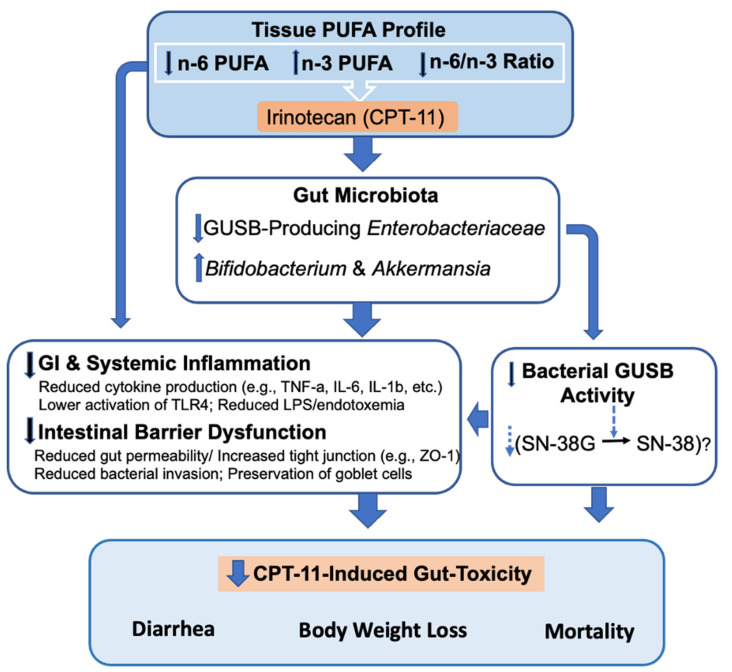
Proposed mechanisms for the preventive effects of the decreased tissue n-6/n-3 ratio on the CPT-11-induced gut toxicities. Diagram illustrating that elevated tissue omega-3 PUFA status with a reduced tissue n-6/n-3 PUFA ratio alters gut microbiome composition and functions, exerts anti-inflammatory and mucosal protective effects, and additionally, reduces the abundance of GUSB-producing bacteria, GUSB activity and potentially the conversion of inactive SN-38G to toxic SN-38. These alterations together with other gut-microbiota-independent mechanisms reduce mucosal injuries, mucosal inflammation, goblet cell dysfunction, impairment in the gut barrier, and systemic endotoxemia, resulting in the prevention of CPT-11-induced gut toxicities (bodyweight loss, late-onset diarrhea, and bloody diarrhea, and death).

## Data Availability

16S rRNA gene sequencing data (raw FASTQ files, QIIME2 bioinformatics analysis results, and PICRUST2 data) generated in this study have been made publicly available in Figshare (DOI: https://doi.org/10.6084/m9.figshare.13606988 (accessed on 15 February 2022); DOI: https://doi.org/10.6084/m9.figshare.13607204 (accessed on 15 February 2022)).

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
