# Peer review of "Decreased Tissue Omega-6/Omega-3 Fatty Acid Ratio Prevents Chemotherapy-Induced Gastrointestinal Toxicity Associated with Alterations of Gut Microbiome"

_ijms, 2022, doi:10.3390/ijms23105332_

Round 1
Reviewer 1 Report
Kaliannan K et al reported that the ratio of n-6/n-3 PUFA is a key factor for the control of chemotherapy-induced gastrointestinal toxicity. The authors concluded that the mechanisms involve gut microbiome alteration with different enzymatic activity to produce toxic metabolite. Thier findings are interesting and originality is high but the association of the intestinal toxicity and enzyme activity is obscure. This reviewer questions whether the phenotypes are ascribed to bacterial enzaymatic activity change or not. Belows are my specific comments.
Major issues:
As n-3 PUHAs exert anti-inflammatory activities by directly acting on host immune cells such as neutrophils, the association of altered GUSB activity and improved GIT is obscure. Did the author examine the amount of SN-38G and SN38 between WT-CPT11 and fat1-CPT-11 mice?
Fig. 2M showed that n-3 PUFAs had little effects on the GUSB activity in the steady state. The questions are (i) how CPT-11 change microbiota composition to increase GUSB activity, and (ii) how n-3 PUFAs can control CPT-11-induced these alterations.
Please clarify the molecules of n-6 and n-3 PUFAs which you examined as total amount with gas chromatography.
Fig 3I and 3J: The authors showed that the gene expression level of MUC2 was reduced in WT-CPT11 group, while that of MUC4 was not. Please give interpretation for these results more in detail.
Supplementary Materials and Methods should be moved to the main text.
Minor issue:
Line 116: The sentence is not reflected by the data of Fig. S1F. Also, please clarify the phenotypes of GI symptoms exactly.
Figure legend of Fig. S1: Representative picture of bloody diarrhea (red circle) is appeared on (F), but not (E).
Line 217 and 218: The authors should describe the results of Fig. S2A-D and Fig. S2E-F, precisely.
Fig. 3F: The data shown here are not convincing. Please show the representative pictures.
Fig. 3G: The authors stained the tissues with different types of reagents: Alcian Blue and PAS. They are both for the counting the goblet cells? Or, for any functional differences of goblet cells?
Fig. S2H and S2I: The data labelling may be wrong. The data shown here are not convincing. Which cells are positive for MUC2 or MUC4?
Fig. 4D: Description about Fig. 4D is missed in the main text.
Author Response
Point-by-point response
Dear Editor,
Please find enclosed our revised manuscript entitled “Decreased Tissue Omega-6/Omega-3 Fatty Acid Ratio Prevents Chemotherapy-Induced Gastrointestinal Toxicity Associated with Alterations of Gut Microbiome”. We thank you and the reviewers for your careful review of our manuscript and helpful comments. We have fully addressed the reviewers’ comments point-by-point as follows and made changes accordingly to the manuscript as highlighted in the text.
Reviewer 1:
Comments and Suggestions for Authors
Kaliannan K et al reported that the ratio of n-6/n-3 PUFA is a key factor for the control of chemotherapy-induced gastrointestinal toxicity. The authors concluded that the mechanisms involve gut microbiome alteration with different enzymatic activity to produce toxic metabolite. Thier findings are interesting and originality is high but the association of the intestinal toxicity and enzyme activity is obscure. This reviewer questions whether the phenotypes are ascribed to bacterial enzaymatic activity change or not. Belows are my specific comments.
Major issues:
As n-3 PUHAs exert anti-inflammatory activities by directly acting on host immune cells such as neutrophils, the association of altered GUSB activity and improved GIT is obscure. Did the author examine the amount of SN-38G and SN38 between WT-CPT11 and fat1-CPT-11 mice?
Thanks for asking this important question. Acutely, omega-3 PUFAs exert anti-inflammatory effects through multiple mechanisms, mainly by modulating the production of lipid mediators and cytokines (PMID: 18751910) and thereby directly or indirectly acting on immune cells. Our recent studies have revealed a novel mechanism for the anti-inflammatory effects of omega-3 PUFAs by modulation of gut microbiota, which leads to reduction of LPS production and intestinal permeability and consequently suppression of metabolic endotoxemia (references 18,19). In this study, we found that increased tissue omega-3 levels with a decreased omega-6/omega-3 PUFA ratio could profoundly reduce the CPT-11-induced GI toxicity with a significant change in gut microbiota, especially a reduction of GUSB-producing bacteria. Certainly, the anti-inflammatory effects by either modulation of gut microbiota and/or through other mechanisms may be a main contributor to the observed protection against the CPT-11 toxicity. Nevertheless, our finding that a reduction of GUSB-producing microbes and GUSB enzyme activity in the gut of mice with high omega-3 and low n-6/n-3 ratio is important for us to understand the reduced GI toxicity of CPT-11 in these animals, as bacterial GUSB activity, a key factor for CPT-11 metabolism (conversing SN-38G to SN-38) in the gut, is well known to be positively related to CPT-11-induced GI toxicity. On this basis, we think that it is feasible to interpret that the alterations of gut microbiota are critical for the protective effects on CPT-11 induced GI toxicity due to their general anti-inflammatory effects as well as their suppressing effect on GUBS activity and CTP-11 metabolism in the gut.
We agree that measurement of the amounts of SN-38G and SN38 between WT-CPT11 and fat1-CPT-11 mice can provide more direct evidence for the involvement of PCT-11 metabolites generated by gut microbes. Because it has been well established that bacterial GUSB activity is the key factor for CPT-11 metabolism (conversing SN-38G to SN38) in the gut and closely associated with CPT-11 GI toxicity, our measurements were focused on the abundance of GUSB-producing gut microbiome and GUSB activity. Given the limited amount of each sample, we did not perform the measurement of SN-38G and SN38 then, and unfortunately, we don’t have any samples for this measurement now. We think that the GUSB data presented (without SN38 data) is sufficient to reflect the role of gut microbiota and CPT-11 metabolism in the GI toxicity of CPT-11 and the protective effects of n-3PUFAs.
To better outline the potential mechanisms based on the data we have, we have revised the summary of the results and Figure 8 by emphasizing the anti-inflammatory and mucosal protective effects other than the changes in GUSB-SN-38 resulting from the alterations of gut microbiota.
Fig. 2M showed that n-3 PUFAs had little effects on the GUSB activity in the steady state. The questions are (i) how CPT-11 change microbiota composition to increase GUSB activity, and (ii) how n-3 PUFAs can control CPT-11-induced these alterations.
Several animal studies have shown that there are clear shifts in the microbial composition after CPT-11 treatment (PMID: 17202590) and suggest that host factors play a key role in microbiome changes in response to CPT-11 administration. The following two host factors could cause CPT-11-induced gut microbiota changes:
1) Increased proinflammatory cytokines. Several recent studies have demonstrated that CPT-11 triggers the innate immune response to cause the secretion and release of proinflammatory cytokines such as interleukin (IL)-18, IL-1β, IL-6, and tumor necrosis factor-α (PMID: 28409562/23628721). The increase in the level of proinflammatory cytokines accelerates the discharge of mucin stored in goblet cells and thereby induce vacuole formation, which further influences intestinal microbial ecocline by reducing the number of adhesion sites and decreasing nutrition. These changes cause a reduction in the number of symbiotic bacteria (e.g., Bifidobacterium spp.) and an increase in the number of opportunistic pathogens (e.g., Escherichia coli) (PMID: 23628721).
2) Alteration of gut microbiome-regulating factors (e.g. IAP). Initial production of SN-38 by GUSP-producing commensal gut bacteria (e.g., Clostridium clusters XIVa and IV; PMID: 18537837/ 22364273) or pro-inflammatory gut microbial environment (reference 18) leads to changes of the host factors (e.g., Intestinal Alkaline Phosphatase (IAP); PMID: 20947883 and reference 18) that can regulate the gut microbiome composition. The CPT-11 metabolites-induced gut mucosal injury/inflammation further down regulate IAP (PMID: 17233117) activity and other gut-microbiome regulating host factors, which in turn, causes the gut Enterobacteriaceae expansion. A “vicious cycle” may be formed because CPT-11 might further aggravate this disturbed status instead of readdressing it. This inharmonious state has been called as “microbiota-host-CPT-11 axis” (PMID: 34722328).
In contrast, elevated tissue levels of n-3 PUFAs are effective in reducing proinflammatory cytokines and upregulating the gut-microbiome-regulating factor IAP. Our studies have shown an attenuation of colonic markers of inflammation (e.g., TNF-α) and a significant improvement of chemical-induced colitis in fat-1 mice with elevated n-3 PUFA levels (PMID: 16847262 /20655721). Furthermore, we have shown that elevated tissue n-3PUFAs enhance intestinal production and secretion of intestinal alkaline phosphatase (IAP) (reference 18), which induces changes in the gut bacteria composition with a decrease in the members of phylum proteobacteria (e.g., Enterobacteriaceae), a major GUSB-producing bacterial group (reference 37). In addition, resolvin E1, a specialized pro-resolving mediator synthesized from n-3 PUFA, has recently been shown to significantly upregulate the expression of IAP, which is critical for the maintenance of bacterial homeostasis (PMID: 20660763). In this context, it is conceivable that CPT-11 treatment may cause a change in gut microbe composition to enhance GUSB activity by an increase in proinflammatory cytokines and a decrease in the expression and activity of IAP, while n-3PUFAs can exert opposing effects on inflammation, IAP activity and gut microbiota to decrease GUSB activity. We have added these points in the discussion.
Please clarify the molecules of n-6 and n-3 PUFAs which you examined as total amount with gas chromatography.
The fatty acids examined as total n-6 PUFA with GC include: Linoleic acid (C18:2n6), Gamma-linolenic acid (C18:3n6), Eicosadienoic acid (C20:2n6), Dihomo-gamma-linolenic acid (C20:3n6), Arachidonic acid (C20:4n6), Docosadienoic acid (C22:2n6), Adrenic acid (C22:4n6) and Docosapentaenoic acid (22:5n6).
The fatty acids examined as total n-3 PUFA with GC include: α-Linolenic acid (C18:3n3), Eicosatrienoic acid (C20:3n3), Eicosapentaenoic acid (C20:5n3), Docosapentaenoic acid (C22:5n3), and Docosahexaenoic acid (C22:6n3).
These details have been added to the manuscript.
Fig 3I and 3J: The authors showed that the gene expression level of MUC2 was reduced in WT-CPT11 group, while that of MUC4 was not. Please give interpretation for these results more in detail.
Mucin genes (MUC1-MUC17) are regulated by cytokines, bacterial products, and growth factors. MUC2 gene encodes secreted gel-forming mucin (PMID: 33883940). MUC2 is the predominant structural component of the intestinal mucus layer, is exclusively and abundantly expressed by goblet cells in the colon (PMID: 7926500). MUC4 gene encodes transmembrane mucins and is expressed in colon (PMID: 33883940). After synthesis, MUC2 is secreted into the lumen and forms a protective mucus gel layer that acts as a selective barrier to protect the epithelium from mechanical stress and noxious agents. Several recent studies have indicated that CPT11 triggers the innate immune response to cause the secretion and release of proinflammatory cytokines such as IL-6, and tumor necrosis factor-α (PMID: 28409562/23628721), which in turn, accelerates the discharge of mucin stored in goblet cells (PMID: 34722328).
In our study, decreased goblet cells (fig 3G) and MUC2 gene expression (fig 3I) in WT+CPT-11 mouse colon tissues collected at day 11 may indicate the loss of mucins from the mucosal surface, which may contribute to the onset of diarrhea induced by CPT-11, and depletion of mucin stores may result in loss of integrity (fig 3E) to the mucus barrier during mucositis. Also, lower expression of MUC2 and higher levels of markers of colonic inflammation (fig 3C-D and 3F and Fig. S2A-C) in WT+CPT-11 mice are consistent with the fact that expression of MUC2 is inversely correlated with the severity of inflammation (PMID: 9722984/reference 43).
This information has been added to the Discussion.
Supplementary Materials and Methods should be moved to the main text.
As requested, we have moved the Supplementary Materials and Methods to the main text
Minor issue:
Line 116: The sentence is not reflected by the data of Fig. S1F. Also, please clarify the phenotypes of GI symptoms exactly.
Mice were examined daily for signs of gastrointestinal toxicity (GIT), including GI symptoms: diarrhea (fecal staining of the skin, loose, watery stool) and bloody diarrhea (black sticky stool or frank blood) and changes in gross appearance (e.g., Hunched posture).
Fig. S1F has two pictures in the manuscript of the first submission. The first picture of Fig. S1F indicates bloody diarrhea (GI symptom). In the revised manuscript, the 2nd picture in Fig. S1F was separated and presented as Fig. S1G (gross appearance).
Figure legend of Fig. S1: Representative picture of bloody diarrhea (red circle) is appeared on (F), but not (E).
Thank you. S1F has two pictures in the manuscript of first submission. The first picture (with red circle) of Fig. S1F indicates bloody diarrhea (GI symptom). In the revised manuscript, the 2nd picture in Fig. S1F was separated and presented as Fig. S1G (gross appearance). The changes were made for Fig. S1 legend and in main manuscript accordingly.
Line 217 and 218: The authors should describe the results of Fig. S2A-D and Fig. S2E-F, precisely.
The results of Fig. S2A-D and Fig. S2E-F are described precisely as follows:
We found that decreased n-6/n-3 ratio in the FAT-1 mice treated with CPT-11 is associated with a reduction of bacterial invasion into the mucosal epithelium as measured by gram staining (Fig. 3A) and down-regulated toll-like receptor 4 (TLR 4) expression (Fig. 3B), compared to WT+CPT-11 group. Likewise, FAT-1+CPT-11 group showed a decrease in markers of inflammation such as nuclear factor NF-kappa-B p65 subunit expression (Fig. 3C), TNF-α (Fig. 3D), IL-1β, IL-6, and MCP-1 (Fig. S2A-C)] and increase in the marker of anti-inflammation [IL-10 (Fig S2D)]. In addition, reduced bacterial invasion and inflammation were associated with improvements in the markers of colonic barrier integrity such as zonulin 1 (ZO-1) (Fig. 3E), occludin, and claudin-1tight junction protein expressions (Fig. S2E-F), as well as serum LPS levels (Fig. S2G), compared to WT+CPT-11 group.
We have added these descriptions to the Results 2.3.
Fig. 3F: The data shown here are not convincing. Please show the representative pictures.
Actually, Fig.3F shows representative pictures (not data). We suspect that the reviewer might mean Fig. 3H (data about the goblet cells), if so, representative pictures for the data are already shown in Fig.3D.
Fig. 3G: The authors stained the tissues with different types of reagents: Alcian Blue and PAS. They are both for the counting the goblet cells? Or, for any functional differences of goblet cells?
Alcian Blue (AB) staining was used for the counting of the goblet cells. Both AB and Periodic Acid – Schiff (PAS) staining were used to identify any functional differences of goblet cells. AB stains blue for acid mucins. PAS stains purple for a mixture of neutral and acid mucins, and red/magenta color for neutral mucins alone (PMID: 26813339). The examination of tissues with both AB and PAS might indicate changes in the distribution or pattern of expression of neutral and acid mucins, which are indicative of certain pathological conditions (PMID: 19705025). For example, a spontaneous-colitis was observed in muc2 knock-out mice with significantly reduced acid and neutral mucins indicated by AB/PAS staining (PMID: 26813339). Relevant details about AB and PAS staining have been included in the methods, results and discussion sections of the revised manuscript.
Fig. S2H and S2I: The data labelling may be wrong. The data shown here are not convincing. Which cells are positive for MUC2 or MUC4?
Thank you for finding the wrong labelling. We agree that the immunohistochemical data for MUC2 and MUC4 are not convincing although we expected significant changes in MUC 2 positive goblet cells. So, we removed Fig. S2H and S2I and relevant results associated with them in the revised manuscript.
Fig. 4D: Description about Fig. 4D is missed in the main text.
Thank you for your careful review. We added a description about Fig. 4D in the revised manuscript as requested.
Reviewer 2 Report
Reviewers' comments:
In the present paper, Kaliannan et al. found elevated tissue n-3 PUFA reduces CPT-11-induced weight loss, bloody diarrhea, gut pathological changes, and mortality. Moreover, the author also found the gut microbiome is involved in CPT-11-induced gastrointestinal toxicity. The research is overall significant and the manuscript is well-written. There are several minor suggestions that the authors would like to address to improve the manuscript and they are as follows:
1. The author should measure the colonic level of SN-38 and SN-38G in WT vs Fat-1 mice as well as CO vs FO mice.
2. The microscopic images of Fig.2 N and Fig.5 L are blurry. Please replace them with high-quality images.
3. In addition to neutrophils, are there other key pro-inflammatory immune cells (macrophage, Th17 cells) involved in CPT-11-induced gastrointestinal toxicity? What is the effect of elevated tissue n-3 PUFA on other critical pro-inflammatory immune cells?
4. The figure legend of Fig.2 N and Fig.2 O are incorrect.
Author Response
Point-by-point response
Dear Editor,
Please find enclosed our revised manuscript entitled “Decreased Tissue Omega-6/Omega-3 Fatty Acid Ratio Prevents Chemotherapy-Induced Gastrointestinal Toxicity Associated with Alterations of Gut Microbiome”. We thank you and the reviewers for your careful review of our manuscript and helpful comments. We have fully addressed the reviewers’ comments point-by-point as follows and made changes accordingly to the manuscript as highlighted in the text.
Reviewer 2:
Comments and Suggestions for Authors:
In the present paper, Kaliannan et al. found elevated tissue n-3 PUFA reduces CPT-11-induced weight loss, bloody diarrhea, gut pathological changes, and mortality. Moreover, the author also found the gut microbiome is involved in CPT-11-induced gastrointestinal toxicity. The research is overall significant and the manuscript is well-written. There are several minor suggestions that the authors would like to address to improve the manuscript and they are as follows:
- The author should measure the colonic level of SN-38 and SN-38G in WT vs Fat-1 mice as well as CO vs FO mice.
We agree that measurement of the amounts of SN-38G and SN38 between WT-CPT11 and fat1-CPT-11 mice can provide more direct evidence for the involvement of PCT-11 metabolites generated by gut microbes. Because it has been well established that bacterial GUSB activity is the key factor for CPT-11 metabolism (conversing SN-38G to SN38) in the gut and closely associated with CPT-11 GI toxicity, our measurements were focused on the abundance of GUSB-producing gut microbiome and GUSB activity. Given the limited amount of each sample, we did not perform the measurement of SN-38G and SN38 then, and unfortunately, we don’t have any samples for this measurement now. We think that the GUSB data presented (without SN-38 data) is sufficient to reflect the role of gut microbiota and CPT-11 metabolism in the GI toxicity of CPT-11 and the protective effects of n-3 PUFAs.
To better outline the potential mechanisms based on the data we have, we have revised the summary of results and Figure 8 by emphasizing the anti-inflammatory and mucosal protective effects other than the changes GUSB-SN-38 resulting from the alterations of gut microbiota.
- The microscopic images of Fig.2 N and Fig.5 L are blurry. Please replace them with high-quality images.
Acutely, these images are quite clear. The blurry look may be due to the conversion issue in this version.
- In addition to neutrophils, are there other key pro-inflammatory immune cells (macrophage, Th17 cells) involved in CPT-11-induced gastrointestinal toxicity? What is the effect of elevated tissue n-3 PUFA on other critical pro-inflammatory immune cells?
Thanks for asking this question.
Yes, other key pro-inflammatory immune cells such as macrophages and Th17 cells have been shown to be involved in CPT-11-induced gastrointestinal toxicity. Also, elevated tissue n-3 PUFAs have been shown to have beneficial effects on these critical pro-inflammatory immune cells in various disease contexts as outlined below although we did not examine these cells in this study.
Role of Macrophages
CPT-11 enhanced tumor necrosis factor-α (TNF-α) and interleukin-6 (IL-6) levels and inducible nitric oxide synthase (iNOS) gene expression, associated with an increase in the total number of macrophages and degranulated mast cells in the small intestine segments and caused significant weight loss (PMID: 28388962) in Swiss mice with CPT-11-induced intestinal mucositis.
Role of Th17 cells
1) Fernandes C. et al showed that CPT-11-induced intestinal mucositis caused an accumulation of Tregs and Th17 cells over time in lamina propria of C57BL/6 mice (PMID: 29307857). A higher number of Th17 cells on the fifth and seventh days of mucositis development were detected. Treg depletion exacerbated intestinal damage, diarrhea, neutrophil infiltration, and animal mortality, despite a reduction in Th17 cell number. The frequency of other Th cells increased and was positively correlated with neutrophil infiltration. Tregs showed a negative correlation with neutrophils and the frequency of non-regulatory Th cells. In conclusion, Tregs are important in the control of intestinal damage induced by irinotecan, and their depletion showed a deleterious effect on IM. Activation of Tregs cells appears to be a compensatory mechanism for intestinal inflammation.
2) Th17 cells mediate the conversion of the immune system to a pro-inflammatory state by secreting immunostimulatory cytokines (e.g., IL-17) (PMID: 32580023) or by activating and recruiting neutrophils (PMID: 23157335). The lamina propria of germ-free mice (GFM) lacks these pro-inflammatory Th17 cells. It has been shown that GFM was resistant to CPT-11-indiced GIT (PMID: 16489087), and it is possible that this resistance could mainly be due to lack of pro-inflammatory Th17 cells in the intestine of GFM.
Effect of elevated tissue n-3 PUFA on other critical pro-inflammatory immune cells
1) In fat-1 mice with elevated tissue n-3 PUFAs, specific reactivity of lymphoid elements in the intestine, CD3(+) T cells, CD4(+) T helper cells, and macrophages from colonic lamina propria were quantified in a colitis-associated colon cancer model. Comparison of 3-day versus 2-week recovery time points revealed that fat-1 mice exhibited decreased CD3(+), CD4(+) T helper, and macrophage cell numbers per colon as compared with WT mice (PMID: 18483285).
2) During colon inflammation, Th17 cells and immunosuppressive regulatory T cells (Treg) are thought to play promotive and preventative roles, respectively. Results from fat-1 mice with elevated tissue n-3 PUFAs showed that n-3 PUFA can modulate the colonic mucosal microenvironment to suppress Th17 cell accumulation and inflammatory damage following the induction of chronic colitis (PMID: 22131549). Similarly, antagonizing arachidonic acid-derived eicosanoids with elevated tissue n-3 PUFAs reduces inflammatory Th17 and Th1 cell-mediated inflammation and colitis severity (PMID: 25136149).
Although we did not investigate these cells in this study, we have now mentioned them in the discussion section.
- The figure legend of Fig.2 N and Fig.2 O are incorrect.
Thank you. We have corrected the figure legend of Fig.2N and Fig. 2O.
Round 2
Reviewer 1 Report
The authors adequately addressed all the points. I am satisfied with their responses.